# Differentiable Logic Machines

**Matthieu Zimmer**  *matthieu.zimmer@huawei.com*
*Huawei Noah's Ark Lab, London, United Kingdom.*

**Xuening Feng**  *cindyfeng2019@sjtu.edu.cn*
*UM-SJTU Joint Institute, Shanghai Jiao Tong University, Shanghai, China.*

**Claire Glanois**  *clgl@itu.dk*
*IT University of Copenhagen, Copenhagen, Denmark.*

**Zhaohui Jiang**  *jiangzhaohui@sjtu.edu.cn*
*UM-SJTU Joint Institute, Shanghai Jiao Tong University, Shanghai, China.*

**Jianyi Zhang**  *zhangjy97@sjtu.edu.cn*
*UM-SJTU Joint Institute, Shanghai Jiao Tong University, Shanghai, China.*

**Paul Weng**  *paul.weng@sjtu.edu.cn*
*UM-SJTU Joint Institute, Shanghai Jiao Tong University, Shanghai, China.*

**Dong Li**  *lidong106@huawei.com*
*Huawei Noah's Ark Lab, China.*

**Jianye Hao**  *haojianye@huawei.com*
*Huawei Noah's Ark Lab, China. School of Computing and Intelligence, Tianjin University, China.*

**Wulong Liu**  *liuwulong@huawei.com*
*Huawei Noah's Ark Lab, Canada.*

**Reviewed on OpenReview:** *https://openreview.net/forum?id=mXfkKtu5JA*

## Abstract

The integration of reasoning, learning, and decision-making is key to build more general artificial intelligence systems. As a step in this direction, we propose a novel neural-logic architecture, called *differentiable logic machine* (DLM), that can solve both inductive logic programming (ILP) and reinforcement learning (RL) problems, where the solution can be interpreted as a first-order logic program. Our proposition includes several innovations. Firstly, our architecture defines a restricted but expressive continuous relaxation of the space of first-order logic programs by assigning weights to predicates instead of rules, in contrast to most previous neural-logic approaches. Secondly, with this differentiable architecture, we propose several (supervised and RL) training procedures, based on gradient descent, which can recover a fully-interpretable solution (i.e., logic formula). Thirdly, to accelerate RL training, we also design a novel critic architecture that enables actor-critic algorithms. Fourthly, to solve hard problems, we propose an incremental training procedure that can learn a logic program progressively. Compared to state-of-the-art (SOTA) differentiable ILP methods, DLM successfully solves all the considered ILP problems with a higher percentage of successful seeds (up to 3.5×). On RL problems, without requiring an interpretable solution, DLM outperforms other non-interpretable neural-logic RL approaches in terms of rewards (up to 3.9%). When enforcing interpretability, DLM can solve harder RL problems (e.g., *Sorting*, *Path*) than other interpretable RL methods. Moreover, we show that deep logic programs can be learned via incremental supervised training. In addition to this excellent performance, DLM can scale well in terms of memory and computational

time, especially during the testing phase where it can deal with much more constants ($>2\times$) than SOTA.

# 1 Introduction

Following the successes of deep learning and deep reinforcement learning, a research trend (Dong et al., 2019; Jiang & Luo, 2019; Manhaeve et al., 2018), whose goal is to combine reasoning, learning, and decision-making into one architecture has become very active. This research may unlock the next generation of artificial intelligence (AI) (Lake et al., 2017; Marcus, 2018). Simultaneously, a second research trend has flourished under the umbrella term of *explainable AI* (Barredo Arrieta et al., 2020). This trend is fueled by the realization that solutions obtained via deep learning-based techniques are difficult to understand, debug, and deploy. Notably, interpretability is crucial in high-stake domains (e.g., autonomous driving), where the decisions made by a trained model should be understandable.

In this context, neural-logic approaches (see Section 2) have been proposed to integrate reasoning and learning, notably via first-order logic and neural networks. Recent works have demonstrated good achievements by using differentiable methods to learn a logic program (Evans & Grefenstette, 2018) or by applying a logical inductive bias to create a neural-logic architecture (Dong et al., 2019). The latter approach currently obtains the best performance at the cost of interpretability, while the former can yield an interpretable solution, but at the cost of scalability. Therefore, one important research problem regards the design of an efficient method with good performance and better scalability while preserving interpretability. The main motivation of our work is to provide such algorithm.

In this paper, we propose a novel neural-logic architecture (see Section 3 for background notions and Section 4 for our proposition) that offers a better tradeoff in terms of interpretability vs. performance and scalability. This architecture defines a continuous relaxation over first-order logic expressions defined on input predicates. In contrast to most previous approaches (see Section 2), one key idea is to assign learnable weights on predicates instead of template rules, which allows for a much better scalability. This architecture can be trained both in the supervised learning (SL) and reinforcement learning (RL) settings for which we introduce several training techniques to find interpretable solutions. Notably, to accelerate the RL training, we propose an adapted critic to train our architecture in an actor-critic scheme. In addition, to help scale further the approach, we propose an incremental training methodology that can be applied to both SL and RL training.

We experimentally compare our proposition with previously-proposed neural-logic architectures in both the SL and RL settings on inductive logic programming (ILP) and RL tasks respectively (see Section 5). Our architecture can achieve state-of-the-art (SOTA) performances in both ILP and RL tasks while maintaining interpretability and achieving better scalability. More precisely, our proposition is superior to all considered interpretable methods in terms of success rates, computational time, and memory consumption. Compared to non-interpretable ones, our method compares favorably, but can find fully-interpretable solutions (i.e., logic programs) that are faster and use less memory during the testing phase.

The contributions of this paper can be summarized as follows: (1) a novel neural-logic architecture that can produce an interpretable solution and that can scale better than SOTA methods, (2) several training algorithms to obtain an interpretable solution, and (3) a thorough empirical evaluation in both RL and ILP tasks. Hence, to the best of our knowledge, our approach is the first neural-logic method able to output at the end of training a fully-interpretable solution for complex tasks like *Path* or *Blocksworld*.

# 2 Related Work

The literature aiming at integrating reasoning, learning, and possibly decision-making is very rich. Our work is related to statistical relational AI (De Raedt et al., 2016), which aims at combining relational reasoning and learning. However, a key difference is that our focus is to learn a logic program, although we have a probabilistic interpretation of predicate evaluations. Our work is also related to relational RL (Džeroski et al., 2001; Tadepalli et al., 2004; van Otterlo, 2012), whose goal is to combine RL with *First-Order Logic* (FOL) representation. To the best of our knowledge, such approach does not scale as well as those resorting

to neural networks. Thus, the investigation of neural approaches to tackle this integration has become very active in recent years (de Raedt et al., 2020; d'Avila Garcez et al., 2019; Besold et al., 2017). For space reasons, we focus on the recent work closest to ours below.

**(Differentiable) ILP and their extensions to RL**  ILP (Muggleton, 1991; Cropper et al., 2020) aims to extract lifted logical rules from examples. Since traditional ILP systems can not handle noisy, uncertain or ambiguous data, they have been extended and integrated into neural and differentiable frameworks. For instance, Evans & Grefenstette (2018) proposed $\partial$ILP, a model based on a continuous relaxation of the logic reasoning process, such that the parameters can be trained via gradient descent, by expressing the satisfiability problem of ILP as a binary classification problem. This relaxation is defined by assigning weights to templated rules. Jiang & Luo (2019) adapted $\partial$ILP to RL problems using vanilla policy gradient. Despite being interpretable, this approach does not scale well in terms of both memory and computation, which is notably due to how the relaxation is defined.

Payani & Fekri (2019b) proposed differentiable Neural Logic ILP (dNL-ILP), another ILP solver where in contrast to $\partial$ILP, weights are placed on predicates like in our approach. Their architecture is organized as a sequence of one layer of neural conjunction functions followed by one layer of neural disjunction functions to represent expressions in Conjunctive Normal Form (CNF) or Disjunctive Normal Form (DNF), which provides high expressivity. In this architecture, conjunctions and disjunctions are defined over all predicates of any arity in contrast to DLM. Payani & Fekri (2019b) did not provide any experimental evaluation of dNL-ILP on any standard ILP benchmarks. But, in our experiments, our best effort to evaluate it suggests that dNL-ILP performs worse than $\partial$ILP. We believe this is due to the too generic form imposed on the logic program to be learned. Payani & Fekri (2020) extended their model to RL and showed that initial predicates can be learned from images if sufficient domain knowledge under the form of auxiliary rules is provided to the agent. However, they do not show that their approach can learn good policies without this domain knowledge.

Another way to combine learning and logic reasoning is to introduce some logical architectural inductive bias, as in Neural Logic Machine (NLM) (Dong et al., 2019). This approach departs from previous ones by learning rules with multilayer perceptrons (MLPs), which prevent this method to provide any final interpretable solution. NLM can generalize and its inference time is significantly reduced compared to $\partial$ILP; by avoiding rule templates as in traditional neural-symbolic approaches, it also gains in expressivity. Our architecture is inspired by NLM, but we use interpretable modules instead of MLPs.

Since the search space of traditional and differentiable ILP methods is often very different, it is difficult to produce a fair comparison. As far as we know, traditional ILP methods often rely on carefully hand-designed and task-specific templates which is less the case with our approach. This difficulty and the difference between these approaches may explain the limited comparative evaluation in past work on ILP (Law et al., 2018; Glanois et al., 2022). Glanois et al. (2022) state that traditional methods are superior to neuro-symbolic ones only when the dataset is small and noise-free. In the RL setting, it is also not clear how to extend non-differentiable methods.

**Other neural-symbolic approaches**  In order to combine probabilistic logic reasoning and neural networks, Manhaeve et al. (2018) proposed DeepProbLog, which extends ProbLog De Raedt et al. (2007), a probabilistic logic language, with neural predicates. While this approach is shown to be capable of program induction, it is not obvious how to apply it for solving generic ILP problems, since partially-specified programs need to be provided.

Another line of work in relational reasoning specifically targets Knowledge-Base (KB) reasoning. Although these works have demonstrated a huge gain in scalability (w.r.t. the number of predicates or entities), they are usually less concerned about predicate invention[1]. Some recent works (Yang et al., 2017; Yang & Song, 2020) extend the multi-hop reasoning framework to ILP problems. The latter work is able to learn more expressive rules, with the use of nested attention operators. In the KB completion literature, a recurrent idea is to jointly learn sub-symbolic embeddings of entities and predicates, which are then used for approximate

---

[1]Typically, rules learned in KB reasoning are chain-like rules (i.e., paths on graphs), which form a subset of Horn clauses: $Q(X, Y) \leftarrow P_1(X, Z_1) \wedge P_2(Z_1, Z_2) \wedge \cdots P_b(Z_{b-1}, Z_b)$.

inference. However, the expressivity remains too limited for more complex ILP tasks and these works are typically more data-hungry.

## 3 Background

We present the necessary notions in ILP and RL. We also recall NLM, since our work is based on it.

### 3.1 Inductive Logic Programming (ILP)

ILP (Muggleton, 1991) refers to the problem of learning a logic program that entails a given set of positive examples and does not entail a given set of negative examples. This logic program is generally written in (a fragment of) FOL.

FOL is a formal language defined with several elements: constants, variables, functions, predicates, and formulas. *Constants* correspond to the objects in the domain of discourse. Let $\mathcal{C}$ denote the set of $m$ constants (e.g., objects). They will be denoted in lowercase (e.g., $o$). *Variables* refer to unspecified constants. They will be denoted in uppercase (e.g., $X$, $Y$, or $Z$). *Functions* allow to denote some constants using other constants (e.g., $s(0)$ may refer to 1). Like previous work, we consider a fragment of FOL without any functions. A *predicate* can be thought of as a relation between constants, which can be evaluated as true $\mathbb{T}$ or false $\mathbb{F}$. A predicate is said to be *b-ary* if it is a relation between $b$ constants. Note that a 0-ary predicate is simply a Boolean value. Predicates will be denoted in uppercase (e.g., $P$ or $Q$). Let $\mathcal{P}$ denote the set of predicates used for a given problem. An *atom* is an $b$-ary predicate with its arguments $P(x_1, \cdots, x_b)$ where $x_i$'s are either variables or constants. For simplicity, we may refer to atoms as predicates when there is no risk of confusion. A *formula* is a logical expression composed of atoms, logical connectives (e.g., negation $\neg$, conjunction $\wedge$, disjunction $\vee$, implication $\leftarrow$), and possibly existential $\exists$ and universal $\forall$ quantifiers.

Since solving an ILP task involves searching an exponentially large space, this problem is generally handled by focusing on formulas of restricted forms, such as a subset of if-then *rules*, also referred to as *clauses*. A *definite clause* is a rule of the form:

$$H \leftarrow A_1 \wedge \ldots \wedge A_k$$

which means that the head atom $H$ is implied by the conjunction of the body atoms $A_1, \ldots, A_k$. *Horn clauses* extend definite rules by allowing $H$ to be possibly negated. More general rules can be defined by allowing logical operations (e.g., disjunction or negation) in the body. A *ground rule* (resp. *ground atom*) is a rule (resp. atom) whose variables have been all replaced by constants.

In ILP tasks, given some *initial predicates* (e.g., for natural numbers, $Zero(X)$ meaning "$X = 0$", $Succ(X, Y)$ meaning "$X = Y + 1$"), and a *target predicate* (e.g., $Even(X)$ meaning "$X$ is even"), the goal is to learn a logical formula defining the target predicate. Expressing directly this target predicate in terms of initial ones may require a very long logical formula, which may therefore be hard to learn. A usual approach is to rely on *predicate invention*, which consists in learning intermediate *auxiliary predicates* with which the target predicate can be defined. Below, we show a simple example of such predicate invention with the created auxiliary predicate $Succ2(X, Y)$ (meaning "$X = Y + 2$"):

$$Even(X) \leftarrow Zero(X) \vee \big(Succ2(X, Y) \wedge Even(Y)\big),$$
$$Succ2(X, Y) \leftarrow Succ(X, Z) \wedge Succ(Z, Y).$$

### 3.2 Reinforcement Learning (RL)

For any finite set $\mathcal{X}$, let $\Delta(\mathcal{X})$ denote the set of probability distributions over $\mathcal{X}$. The *Markov Decision Process* (MDP) model (Bellman, 1957) is defined as a tuple $(\mathcal{S}, \mathcal{A}, T, r, \mu, \gamma)$, where $\mathcal{S}$ is a set of states, $\mathcal{A}$ is a set of actions, $T : \mathcal{S} \times \mathcal{A} \to \Delta(\mathcal{S})$ is a transition function, $r : \mathcal{S} \times \mathcal{A} \to \mathbb{R}$ is a reward function, $\mu \in \Delta(\mathcal{S})$ is a distribution over initial states, and $\gamma \in [0, 1)$ is a discount factor. A (stationary Markov) policy $\pi : \mathcal{S} \to \Delta(\mathcal{A})$ is a mapping from states to distributions over actions; $\pi(a \mid s)$ stands for the probability of taking action $a$ given state $s$. We consider parametrized policies $\pi_{\boldsymbol{\theta}}$ with parameter $\boldsymbol{\theta}$ (e.g., neural networks). The aim in

discounted MDP settings is to find a policy maximizing the expected discounted total reward:

$$J(\boldsymbol{\theta}) = \mathbb{E}_{\mu,T,\pi_{\boldsymbol{\theta}}}\Big[ \sum_{t=0}^{\infty} \gamma^t r(s_t, a_t) \Big], \tag{1}$$

where $\mathbb{E}_{\mu,T,\pi_{\boldsymbol{\theta}}}$ is the expectation w.r.t. distribution $\mu$, transition function $T$, and policy $\pi_{\boldsymbol{\theta}}$. The *state value function* of a policy $\pi_{\boldsymbol{\theta}}$ for a state $s$ is defined by:

$$V^{\boldsymbol{\theta}}(s) = \mathbb{E}_{T,\pi_{\boldsymbol{\theta}}}\Big[ \sum_{t=0}^{\infty} \gamma^t r(s_t, a_t) \mid s_0 = s \Big], \tag{2}$$

where $\mathbb{E}_{T,\pi_{\boldsymbol{\theta}}}$ is the expectation w.r.t. transition function $T$ and policy $\pi_{\boldsymbol{\theta}}$. The *action value function* is defined by:

$$Q^{\boldsymbol{\theta}}(s,a) = \mathbb{E}_{T,\pi_{\boldsymbol{\theta}}}\Big[ \sum_{t=0}^{\infty} \gamma^t r(s_t, a_t) \mid s_0 = s, a_0 = a \Big],$$

and the *advantage function* (Sutton & Barto, 2018) is defined by:

$$A^{\boldsymbol{\theta}}(s,a) = Q^{\boldsymbol{\theta}}(s,a) - V^{\boldsymbol{\theta}}(s).$$

RL (Sutton & Barto, 2018), which is based on MDP, is the problem of learning a policy that maximizes the expected discounted sum of rewards without knowing the transition and reward functions. Policy Gradient (PG) methods constitute a widespread approach for tackling RL problems in continuous or large state-action spaces. They are based on iterative updates of the policy parameter in the direction of a (policy) gradient expressed as:

$$\nabla_{\boldsymbol{\theta}} J(\boldsymbol{\theta}) = \mathbb{E}_{s \sim d^{\pi_{\boldsymbol{\theta}}}(\cdot), a \sim \pi_{\boldsymbol{\theta}}(s)}[A^{\boldsymbol{\theta}}(s,a) \nabla_{\boldsymbol{\theta}} \log \pi_{\boldsymbol{\theta}}(a \mid s)],$$

where the expectation is taken w.r.t. $d^{\pi_{\boldsymbol{\theta}}}(s) = \sum_{s_0} \mu(s_0) \sum_{t=1}^{\infty} \gamma^{t-1} Pr(s_t = s|s_0)$, the discounted state distribution induced by policy $\pi_{\boldsymbol{\theta}}$. Algorithms like REINFORCE (Williams, 1992) that estimate this gradient via Monte Carlo sampling are known to suffer from high variance (due to the stochasticity of the environment and/or policy) Sutton & Barto (2018). To address this issue, actor-critic (AC) schemes (Konda & Tsitsiklis, 2000) have been proposed. In such a framework, both an actor ($\pi_{\boldsymbol{\theta}}$) and a critic (e.g., depending on the AC algorithm, $V^{\boldsymbol{\theta}}$ (Sutton & Barto, 2018; Schulman et al., 2017)or $Q^{\boldsymbol{\theta}}$ (Lillicrap et al., 2016; Mnih et al., 2016), from which $A^{\boldsymbol{\theta}}$ can be estimated) are jointly learned. Using a critic to estimate the policy gradient reduces variance, but at the cost of introducing some bias.

*Proximal Policy Optimization* (PPO) (Schulman et al., 2017) is a SOTA AC algorithm, which optimizes a clipped surrogate objective function $J_{\text{PPO}}(\boldsymbol{\theta})$:

$$\mathbb{E}\Big[ \sum_{t=0}^{\infty} \min(\omega_t(\boldsymbol{\theta}) A^{\bar{\boldsymbol{\theta}}}(s_t, a_t), \text{clip}(\omega_t(\boldsymbol{\theta}), \epsilon) A^{\bar{\boldsymbol{\theta}}}(s_t, a_t)) \Big],$$

where $\bar{\boldsymbol{\theta}}$ is the parameter of the policy that generated the training data, $\omega_t(\boldsymbol{\theta}) = \frac{\pi_{\boldsymbol{\theta}}(a_t|s_t)}{\pi_{\bar{\boldsymbol{\theta}}}(a_t|s_t)}$, the advantage function $A^{\bar{\boldsymbol{\theta}}}$ is estimated with Generalised Advantage Estimation (GAE) (Schulman et al., 2016) and $\text{clip}(\cdot, \epsilon)$ is the function to clip between $[1 - \epsilon, 1 + \epsilon]$. This surrogate objective was motivated as an approximation of that used in Trust Region Policy Optimization (TRPO) (Schulman et al., 2015), which was introduced to ensure monotonic improvement after a policy parameter update. Some undeniable advantages of PPO over TRPO lie in its simplicity and lower computational and sample complexity.

### 3.3 Neural Logic Machine (NLM)

NLMs (Dong et al., 2019) are neural networks designed with a strong architectural inductive bias to solve ILP (and RL) problems. The NLM architecture is composed of MLPs organized in a hierarchical fashion. Its input corresponds to the initial predicates and its output corresponds to the target predicate. The MLPs approximate logic operations and define invented auxiliary predicates based on previously-created or initial

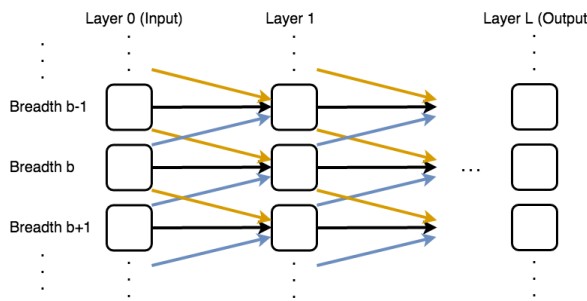
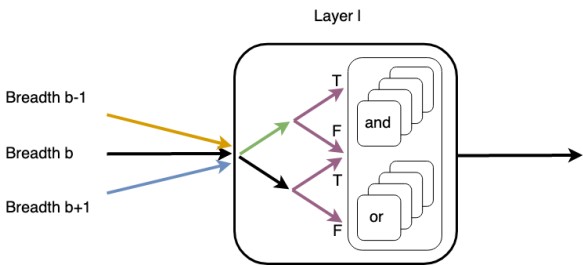

(a) High-level architecture of NLM (and DLM) zoomed in around breadth $b$, where boxes represent computation units (except for layer 0), blue arrows correspond to reduction, and yellow arrows to expansion. Permutation is applied on the inputs of each computation unit.

(b) A DLM computation unit at breadth $b$ and layer $l$. After the NLM operations (i.e., reduction, expansion, and permutation), negation (green arrow) and preservation (violet arrow) with true (T) or false (F) are applied to generate the inputs of the logic modules.

Figure 1: (Left) High-level architecture of NLM (and DLM), (Right) Computation unit in DLM.

predicates. Thus, this architecture simulates forward chaining, a form of reasoning, which computes a target predicate from the truth values of initial predicates via a sequence of invented ones. Training an NLM therefore approximates the inductive definition of logic formulas.

More specifically, an NLM is comprised of learnable computation units organized into $L$ successive layers, each layer having a maximum breadth $B$ (see Fig. 1a). A computation unit at layer $l$ for a given breadth $b$ corresponds to some $b$-ary predicates and is connected to units at the previous and next layers. An NLM processes atoms (i.e., predicates with all its arguments) represented as tensors. For any $b$-ary predicate $P$, its corresponding tensor $\boldsymbol{P}$, denoted in bold, is of order $b$ with shape $[m, \dots, m]$ (recall that $m$ is the number of constants present in the ILP problem). For instance, the tensor representation of a binary ($b = 2$) predicate would be $\boldsymbol{P} \in [0,1]^{m \times m}$. A value $\boldsymbol{P}(i_1, \dots, i_b) \in [0, 1]$ (with $i_j \in \{1, \dots, m\}$ and $j \in \{1, \dots, b\}$) is interpreted as the probability of the truth value of the corresponding grounded atom.

Apart from layer 0, which directly corresponds to the initial predicates, a computation unit at breadth $b \in \{0, \dots, B\}$ and layer $l \in \{1, \dots, L\}$ takes as inputs the $b$-ary predicates of a set $\mathcal{Q}_b^l$ and then outputs a set $\mathcal{P}_b^l$ of newly-invented $b$-ary predicates. For $l > 0$, the set $\mathcal{Q}_b^l$ is obtained from the sets $\mathcal{P}_b^{l-1}$ of the previous layer according to three operations: *expansion*, *reduction*, and *permutation*:

**Expansion:** Any $b$-ary predicate $P$ can be expanded into a $(b+1)$-ary predicate $\hat{P}$, where the last argument does not play any role in its truth value, i.e., $\hat{P}(X_1, \dots, X_{b+1}) := P(X_1, \dots, X_b)$. The set of expanded predicates obtained from $\mathcal{P}_b^{l-1}$ is denoted $\widehat{\mathcal{P}}_b^{l-1}$.

**Reduction:** Any $(b+1)$-ary predicate $P$ can be reduced into a $b$-ary predicate $\check{P}$ by marginalizing out its last argument with an existential (resp. universal) quantifier, i.e., $\check{P}(X_1, \dots, X_b) = \exists X_{b+1} P(X_1, \dots, X_{b+1})$ (resp. $\check{P}(X_1, \dots, X_b) = \forall X_{b+1} \; P(X_1, \dots, X_{b+1})$). Those operations on tensors are performed by a max (resp. min) on the corresponding component, i.e., $\check{\boldsymbol{P}}(i_1, \dots, i_b) = \max_j \boldsymbol{P}(i_1, \dots, i_b, j)$ (resp. $\check{\boldsymbol{P}}(i_1, \dots, i_b)$ $= \min_j \boldsymbol{P}(i_1, \dots, i_b, j)$). The set of reduced predicates obtained from $\mathcal{P}_{b+1}^{l-1}$ is denoted $\check{\mathcal{P}}_{b+1}^{l-1}$.

**Permutation:** Let $\mathbb{S}_b$ be the set of all permutations of $\{1, \dots, b\}$ for $b \in \{1, \dots, B\}$. For a given $b$-ary predicate $P$ and a given permutation $\sigma \in \mathbb{S}_b$, $P_\sigma(X_1, \dots, X_b) := P(X_{\sigma(1)}, \dots, X_{\sigma(b)})$. Permuting arguments allows to build more expressive formulas.

**Example 1.** *In a blocks world environment, consider the Unstack task where the goal is to move all the blocks on the floor. Therefore, a constant in $\mathcal{C}$ is a block. The current configuration of the blocks can be described by the following initial predicates: $IsFloor(X)$ (i.e., true if $X$ is the floor), $On(X, Y)$ (i.e., true if $X$ is on $Y$), and $Top(X)$ (i.e., true if there is no block on $X$). The target predicate $Move(X, Y)$ to be learned indicates which block $X$ could be moved on $Y$ so that the next configuration of the blocks is closer to*

*the goal specified by the Unstack task. A possible solution is:*

$$Move(X, Y) \leftarrow IsFloor(Y) \wedge Top(X) \wedge \exists Z, P(X, Z),$$
$$P(X, Y) \leftarrow On(X, Y) \wedge \exists Z, On(Y, Z),$$

*where $P$ is an invented predicate, which can be formulated thanks to those three previous operations. Indeed, reduction applied on $On$ can yield $Q(Y) = \exists Z, On(Y, Z)$. Applying expansion on it yields $\forall X, \hat{Q}(Y, X) = Q(Y)$. Using permutation $\sigma$ that swaps two arguments gives $\hat{Q}_\sigma(X, Y) = \hat{Q}(Y, X)$. Finally, $P$ can be expressed with $On(X, Y)$ and $\hat{Q}_\sigma(X, Y)$.*

The input predicates of a computation unit at layer $l$ and breadth $b$ are therefore the elements of set $\mathcal{Q}_b^l = \{P_\sigma \mid P \in \mathcal{P}_b^{l-1} \cup \widehat{\mathcal{P}}_{b-1}^{l-1} \cup \widecheck{\mathcal{P}}_{b+1}^{l-1}, \sigma \in \mathbb{S}_b\}$ assuming $\widehat{\mathcal{P}}_0^l = \varnothing$ and $\widecheck{\mathcal{P}}_{B+1}^l = \varnothing$. A computation unit outputs $n_O$ new predicates, which form the elements of set $\mathcal{P}_b^l$ of $b$-ary predicates invented at layer $l$. For each such predicate, a corresponding MLP computes its outputs: its evaluation for any given grounding (i.e., its arguments $o_1, \ldots, o_b$) is computed with the same MLP using as inputs all predicates in $\mathcal{Q}_b^l$ grounded on $o_1, \ldots, o_b$. Thus, the NLM architecture is independent of the number of constants and therefore a trained network can be applied on new instances with any number of constants.

NLM offers an expressive neural network architecture, whose model complexity is controlled by setting the number $L$ of layers, breadth $B$, and number $n_O$ of output predicates of the computation units, in addition to the MLP size. However, by using MLPs, NLM cannot provide an interpretable solution after training.

## 4   Differentiable Logic Machine (DLM)

We present our novel neural-logic architecture, called Differentiable Logic Machines (DLM), which offers a good trade-off between expressivity and trainability. We first discuss its architecture, then present several training methods, and finally explain how to extract interpretable solutions.

### 4.1   Architecture

At a high level, DLM has a similar architecture (Fig. 1a) as NLM (Dong et al., 2019), i.e., it contains computation units organized into a hierarchical fashion with breadth $B$ and $L$ layers. In contrast to NLM, the computation units correspond to soft logic operators, which means that DLM defines a continuous relaxation over FOL programs. Next, we explain the three main innovations in DLM and compare its computational complexity with respect to other related neural-logic methods. In Appendix A, we provide further discussions about DLM, notably its interpretation, expressivity, and implementation details.

**Innovations**   The first two innovations correspond to two novel operations, *negation* and *preservation*, which we introduce to increase expressivity. They are used in addition to the three operations in NLM (expansion, reduction, and permutation) to compute the input predicates of a computation unit. For the third innovation, we replace the MLPs of NLM's computation units by *logic modules* to promote interpretability. We present the two operations next, and then describe the logic modules:

**Negation:** On any set of predicates $\mathcal{P}$, this operation yields the set $\eta(\mathcal{P})$ containing the negation of the predicates in $\mathcal{P}$. On tensor representations, the negation is computed via the involutive function $f(x) = 1 - x$ applied component-wisely.

**Preservation:** To add more flexibility in the architecture, we augment any set of predicates $\mathcal{P}$ with $\mathbb{T}$ or $\mathbb{F}$, which can be seen as a predicate that is always true or false respectively. The rationale for introducing those constant predicates $\mathbb{T}$ and $\mathbb{F}$ is notably to allow a predicate at one layer to be preserved for a later layer (see Example 2 below).

From the set $\mathcal{Q}_b^l$ of input predicates computed with expansion, reduction, and permutation, the application of negation and preservation yields four sets $\mathcal{Q}_b^l \cup \{\mathbb{T}\}$, $\mathcal{Q}_b^l \cup \{\mathbb{F}\}$, $\eta(\mathcal{Q}_b^l) \cup \{\mathbb{T}\}$, and $\eta(\mathcal{Q}_b^l) \cup \{\mathbb{F}\}$. The union of these sets, denoted $\mathcal{R}_b^l$, is used as input of a logic module, which we explain next.

**Logic module:** In DLM, a computation unit (see Fig. 1b) at layer $l$ and breadth $b$ outputs $n_O$ new predicates like in NLM. However, in contrast to NLM, each invented predicate is based on input predicates from $\mathcal{R}_b^l$ and is computed with a (soft) logic operator. For simplicity, we only use *and* and *or*, but other logic operators could be considered.

An *and* (resp. *or*) logic module at any layer $l$ and breadth $b$ outputs a conjunction (resp. disjunction) over $n_A$ terms of $\mathcal{R}_b^l$, which represents a predicate $C_b^l(X_1, \ldots, X_b)$ (resp. $D_b^l(X_1, \ldots, X_b)$). In terms of tensor, a conjunctive predicate $C_b^l(X_1, \ldots, X_b)$ is computed with a fuzzy *and* and is obtained as follows (for $n_A = 2$, which easily extends to $n_A > 2$):

$$\boldsymbol{C}_b^l = \boldsymbol{Q} \odot \boldsymbol{Q}', \tag{3}$$

where $\boldsymbol{Q} = \sum_{P \in \mathcal{R}_b^l} w_P \boldsymbol{P}$, $\boldsymbol{Q}' = \sum_{P' \in \mathcal{R}_b^l} w_{P'} \boldsymbol{P}'$, $\odot$ is the component-wise product, $w_P \in [0, 1]$ and $w_{P'} \in [0, 1]$ are learnable weights for selecting predicates $P$ and $P'$ such that $\sum_{P \in \mathcal{R}_b^l} w_P = 1$ and $\sum_{P' \in \mathcal{R}_b^l} w_{P'} = 1$. Similarly, a disjunctive predicate $D_b^l(X_1, \ldots, X_b)$ is computed with a fuzzy *or* and is defined as follows (for $n_A = 2$, which easily extends to $n_A > 2$):

$$\boldsymbol{D}_b^l = \boldsymbol{Q} + \boldsymbol{Q}' - \boldsymbol{Q} \odot \boldsymbol{Q}'. \tag{4}$$

Each logic module (i.e., each conjunction or disjunction) has its own set of weights $w_P$'s (and $w_{P'}$'s), which are learned as a softmax of parameters $\theta_P$ (and $\theta_{P'}$) with temperature $\tau$ as a hyperparameter:

$$w_P = \frac{\exp(\theta_P/\tau)}{\sum_{P' \in \mathcal{R}_b^l} \exp(\theta_{P'}/\tau)}. \tag{5}$$

**Example 2.** *Assume $n_A = 2$. Consider a blocks world environment with an initial predicate: $On(X, Y)$ (i.e., block $X$ is on $Y$). If it were not initially provided, predicate $Top(X)$ (i.e., no block is on $X$) could be learned thanks to the negation and preservation operations (in addition to reduction and permutation). Indeed, since $Top(X)$ can be expressed as $\mathbb{T} \wedge \neg(\exists Y \, On(Y, X))$, it can be obtained as follows. As $\mathcal{Q}_1^1$ contains a predicate $P(X) = \exists Y \, On(Y, X)$ (by reduction and permutation), an* and *logic module could express $Top(X)$ as a conjunction of $\mathbb{T} \in \mathcal{Q}_1^1 \cup \{\mathbb{T}\}$ (by preservation) and $\neg P \in \eta(\mathcal{Q}_1^1)$ (by negation). Since $\neg P$ is an input predicate in $\mathcal{R}_1^1$, $\mathbb{T}$ (resp. $\mathbb{F}$) can preserve it via* and *(resp.* or*) logic modules for the next layer.*

**Computational Complexity** In one computation unit, the number of parameters grows as $\mathcal{O}(p n_A n_O)$ where $p$ is the number of input predicates of the unit, $n_A$ is the number of atoms used to define an invented predicate, and $n_O$ is the number of output predicates invented by the unit. In comparison with related work (see Section 2), to obtain the same expressivity, the alternative approach $\partial$ILP (Evans & Grefenstette, 2018) (and thus its extension to RL, NLRL (Jiang & Luo, 2019)) would need $\mathcal{O}(\binom{p}{n_A} \times n_O)$ because the weights are defined for all the $n_A$-combinations of $p$ predicates. In contrast, dNL-ILP and NLM would be better with only $\mathcal{O}(p n_O)$. However, the components of dNL-ILP and NLM are not really comparable to those of our model or $\partial$ILP. Indeed, the NLM units are not interpretable, and the dNL-ILP architecture amounts to learning a CNF or DNF formula, where a component of that architecture corresponds to a part of that formula. While expressive, the space of logic program induced in dNL-ILP is much less constrained than that in our architecture, making it much harder to train, as shown in our experiments (see Table 4).

Assuming that $B$ (the maximum arity) is a small constant, the complete DLM network has $O(L n_A n_O{}^2)$ parameters where $L$ is the maximum number of layers. NLM has instead $O(L n_O{}^2)$ parameters. The complexity of a forward pass in DLM is $O(m^B L n_A n_O{}^2)$ where $m$ is the number of constants. For NLM, it is $O(m^B L n_O{}^2)$ (assuming that single-layer perceptrons are used). Once a logic program is extracted (see Section 4.3), we can reduce the forward pass complexity to $O(m^B p)$ where $p$ is the number of elements in the graph. Note that, by construction, $p \leqslant L n_O{}^2 \leqslant L n_A n_O{}^2$.

## 4.2 Training

As a differentiable model, DLM can be trained in both the SL and RL settings. Since any computation unit directly corresponds to a (possibly soft) logic expression, DLM is also independent of the number of

constants: it can be trained on a small number of constants and generalize to a larger number. Next, we discuss supervised training, then present RL training with an actor-critic scheme. In addition, we also present an incremental training approach to tackle hard tasks. Apart from the latter incremental approach, the algorithms are quite standard. We present their pseudo-codes in Appendix A.

### 4.2.1 Supervised Training

SL training can be applied to solve ILP tasks (see Section B.1 for concrete examples): Given a dataset $\mathcal{D}$ expressed with a set of constants $\mathcal{C}$ and the initial predicates in $\mathcal{P}^0 = \bigcup_b \mathcal{P}_b^0$, learn to predict an $r$-ary target predicate $P^T$. DLM can be trained by minimizing a binary cross-entropy loss (e.g., via stochastic mini-batch gradient descent):

$$\mathcal{L}_{\boldsymbol{\theta}}(\boldsymbol{P}^T, \hat{\boldsymbol{P}}^T) = \sum_{o_1,\ldots,o_r \in \mathcal{C}} -\boldsymbol{P}^T(o_1, \ldots, o_r) \log(\hat{\boldsymbol{P}}^T(o_1, \ldots, o_r)) - (1 - \boldsymbol{P}^T(o_1, \ldots, o_r)) \log(1 - \hat{\boldsymbol{P}}^T(o_1, \ldots, o_r)),$$

where $\boldsymbol{\theta}$ corresponds to all the DLM parameters, $\hat{\boldsymbol{P}}^T = \phi_{\boldsymbol{\theta}}(\boldsymbol{\mathcal{P}}^0)$ is the output atom computed by DLM, and $\boldsymbol{\mathcal{P}}^0$ contains the grounded atoms given in $\mathcal{D}$ (i.e., all $P(o_1, \ldots, o_b)$, $\forall b, \forall o_1, \ldots, o_b \in \mathcal{C}, \forall P \in \mathcal{P}_b^0$). As reported in previous neural-logic work, this loss function generally admits many local optima. Besides, there may be global optima that reside in the interior of the continuous relaxation (i.e., not interpretable). Therefore, if we train our model with a standard supervised (or RL training) technique, there is no reason that an interpretable solution would be obtained, even if we manage to completely solve the task.

In order to help training and guide the model towards an interpretable solution, we propose to use three techniques: (a) inject some noise in the softmax defined in (5), (b) decrease temperature $\tau$ during training, and (c) use dropout. For the noise, we use a Gumbel distribution. Thus, the softmax in (5) is replaced by a Gumbel-softmax (Jang et al., 2017):

$$w_P = \frac{\exp((G_P + \theta_P)/\tau)}{\sum_{P' \in \mathcal{R}_b^l} \exp((G_{P'} + \theta_{P'})/\tau)}, \tag{6}$$

where $G_P$ (and $G_{P'}$) are i.i.d. samples from a Gumbel distribution $Gumbel(0, \beta)$. The injection of noise during training corresponds to a stochastic smoothing technique: optimizing by injecting noise with gradient descent amounts to performing stochastic gradient descent on a smoothed loss function. This helps avoid early convergence to a local optimum and find a global optimum. The decreasing temperature favors the convergence towards interpretable solutions. To further help learn an interpretable solution, we additionally use dropout during our training procedure. Dropout helps learn more independent weights, which also promotes interpretability. The scale $\beta$ of the Gumbel distribution and the dropout probability are also decreased with the temperature during learning.

### 4.2.2 Actor-Critic (AC) Training

For RL tasks (see Section 5 for concrete examples), states are assumed to be encoded with a given set $\mathcal{C}$ of constants and the initial predicates in $\mathcal{P}^0 = \bigcup_b \mathcal{P}_b^0$, while action selection can be expressed by an $b$-ary action predicate $P^A$, i.e., $P^A(o_1, \ldots, o_b)$ corresponds to an action. In contrast to ILP, the truth values of grounded atoms (describing a state) may change after performing an action to reflect a new state. The RL problem consists in learning a policy that takes as inputs $\boldsymbol{\mathcal{P}}^0$ (all the grounded atoms of the initial predicates) and that returns an action predicate such that the expected cumulative reward (1) is maximized by selecting actions according to the action predicate.

**Example 3.** *Consider a blocks world environment with the initial predicate $On(X, Y)$ (i.e., block $X$ is on $Y$). The state is defined by all the state atoms (i.e., ground predicates), e.g., keeping only the positive ones, On(a, b) and On(b, floor) could define a state. The action is defined by an action atom, for instance Move(a, floor). The policy network only takes as input the state atoms (represented as tensors) containing their valuations over all the constants and predicts the truth values of the predicate Move over all the constants. A possible resulting next state after taking action Move(a, floor) would be On(b, floor), On(a, floor).*

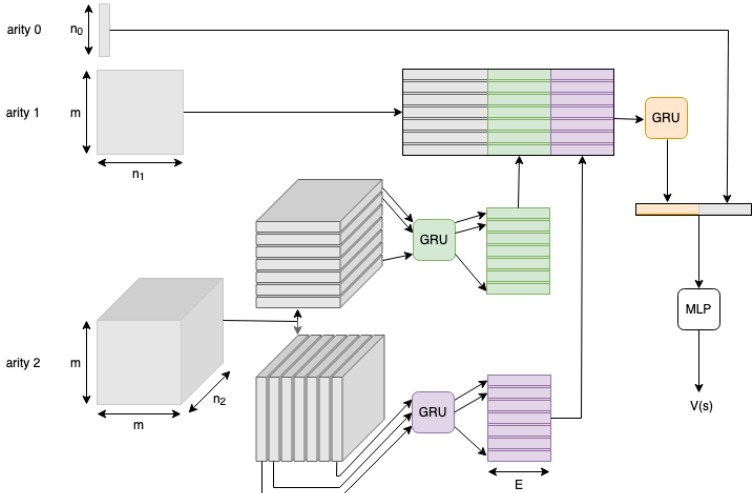

Figure 2: Architecture of our critic, assuming that the max arity in $\mathcal{P}^0$ is 2. On the left are the depicted atoms of arity 0, 1, and 2, where $m = |\mathcal{C}|$, $n_i$ is the number of predicates of arity $i$, and $E$ is the output dimension of a GRU. Each GRU unit sequentially reads slices depicted with black lines. The architecture can be generalized to larger arity by introducing more GRU units. The number of parameters of the critic is independent of the number of constants and only depends on the possible arities and the possible number of predicates in $\mathcal{P}^0$.

To the best of our knowledge, all previous neural-logic works[2] rely on REINFORCE (Williams, 1992) instead of an AC algorithm, which can generally be much more sample-efficient. One reason may be the difficulty of designing a neural network for the critic that can directly receive as inputs $\mathcal{P}^0$ (i.e., the same input given to the policy based on a DLM architecture). To overcome this issue, we propose a recurrent neural network architecture based on Gated Recurrent Unit (GRU) (Cho et al., 2014) for the critic. Once an architecture for the critic is defined, different actor-critic algorithms could be applied. In this work, we use the SOTA AC algorithm, PPO (Schulman et al., 2017). We present next the design of our critic and actor.

**Critic** This GRU-based critic estimates the value (2) of a state described by the atoms in $\mathcal{P}^0$. Recall a GRU unit (Cho et al., 2014) processes an input sequence $x_t$ as follows:

$$z_t = \sigma(W_z x_t + U_z h_{t-1} + b_z),$$
$$\xi_t = \sigma(W_r x_t + U_r h_{t-1} + b_r),$$
$$h_t = z_t \odot h_{t-1} + (1 - z_t) \odot \phi(W_h x_t + U_h(\xi_t \odot h_{t-1}) + b_h),$$

where $z_t$ (resp. $\xi_t$) is the update (resp. reset) gate vector, $h_t$ is the output vector, $\sigma$ (resp. $\phi$) is the sigmoid (resp. tanh) activation function, $\odot$ is the component-wise product, and $W_*, U_*, b_*$ for $* \in \{z, r, h\}$ are learnable parameters.

Our GRU-based critic (see Fig. 2) includes, for each arity $b > 1$ in $\mathcal{P}^0$, $b$ independent GRU units. Each of them reads all the atoms in $\mathcal{P}_b^0$, but focuses on a different component of the $b$-ary predicates. For simplicity, we present the procedure for the first component, the other components are treated in a similar fashion. The GRU unit for the first component processes for each $o \in \mathcal{C}$, the input sequence $x_t$ whose element is defined by $\forall t = (o_2, \ldots o_b) \in \mathcal{C}^{b-1}$, $x_t = \big(P_b(o, o_2, \ldots, o_b)\big)_{P_b \in \mathcal{P}_b^0}$. Its output, taken as the computed last $h_t$, corresponds to $(Q_b^1(o), \ldots, Q_b^E(o)) \in \mathbb{R}^E$ where $E$ is a hyperparameter. Intuitively, the $b$ GRU units compute for each constant a summary (i.e., vector of dimension $b \times E$) of the objects that are in relation with it.

---

[2]Although not discussed in Jiang & Luo (2019), we found in their source code an attempt to apply an AC scheme, but they do so by converting states into images (i.e. by activating the associated pixels in the Cartesian space when a block is present at a specific location), which may not only be unsuitable for some problems, but may also lose information during the conversion and prevent good generalization.

After processing all the atoms of arity $b > 1$, all the combined outputs of these GRU units (i.e., $Q_b^i$'s) in addition to the initial unary predicates are read by another GRU unit, which outputs a vector $(Q_1^1, \ldots, Q_1^E) \in \mathbb{R}^E$. This latter GRU unit takes as inputs $\forall t = o \in \mathcal{C}, x_t \in \mathbb{R}^{N_1}$ where $N_1$ is the total number of $Q_b^i$'s plus the number of initial unary predicates. The computed vector, in addition to any initial 0-ary predicates if any, are given as inputs to an MLP, which outputs the estimated state value. Note that the number of parameters of this critic, like DLM, is independent of the number of constants.

An NLM architecture could also be used for the critic, but we find out that our proposed GRU-based architecture was more efficient in practice. It enables a more expressive aggregation function to compute a value from a set of input tensors. With NLM the reduction of the inputs to a scalar would only use the min-pooling operation across objects, which leads to a too simple aggregation function.

**Actor** The actor's output should ideally correspond to a predicate that evaluates to true for only one action and false for all other actions, which corresponds to a deterministic policy. For instance, in a blocks world domain, the target predicate would be $move(X, Y)$ and would be true for only one pair of objects, corresponding to the optimal action, and false for all other pairs. While not impossible to achieve (at least for certain small tasks), an optimal deterministic policy may involve an unnecessarily complex logic program. Indeed, for instance, for the blocks world domain, in many states, there are several equivalent actions, which the deterministic policy would have to order.

Thus, as done in previous works, we separate the reasoning part and the decision-making part. The reasoning part follows the architecture presented in Fig. 1a, which provides a tensor representing the target predicate corresponding to the actions. A component of this tensor can be interpreted as whether the respective action is good or not. The decision part takes as input this tensor and outputs a probability distribution over the actions using a softmax with fixed low temperature. Note that this temperature used for the actions is different from that used in Equation(6).

### 4.2.3 Incremental Training

Learning an interpretable solution amounts to searching in a discrete space whose size increases exponentially with the size of the solution (i.e., $L$). In complex tasks whose solution requires a logic program with a large depth, the SL or RL training we have just discussed may not be sufficient.

To face this difficulty, we also experimented with a more elaborate training procedure where potentially useful auxiliary predicates are incrementally learned and added to the initial predicates, which can then facilitate finding a good interpretable solution. This is achieved by repeatedly training a fixed DLM architecture with an increasing number of initial predicates. More specifically, we repeat the following training phase. At phase $i$, we apply the SL or RL training method discussed previously on a randomly initialized DLM to obtain an interpretable solution using as initial predicates the set $\mathcal{P}^{(i)}$ of predicates, which includes the predicates of $\mathcal{P}^0 = \bigcup_b \mathcal{P}_b^0$ (i.e., all initial predicates of the problem) in addition to predicates invented in previous phases (if any). Thus, during the first phase, $\mathcal{P}^{(1)} = \mathcal{P}^0$. At the end of a training phase, if the learned interpretable solution solves the problem, the procedure ends. Otherwise, although the obtained solution is imperfect, it may still contain some useful invented predicates. Therefore, we extract (see Section 4.3) the invented predicates from this interpretable solution, which are then added to $\mathcal{P}^{(i)}$ to form $\mathcal{P}^{(i+1)}$ for the next phase.

Intuitively, incremental training stacks many DLM networks (each with $L$ layers) to obtain a large DLM network whose number of layers can be up to the number of phases times $L$, depending on which invented predicates are used. Thus, incremental training can be seen as a training procedure for training a very deep DLM architecture. Note that this procedure increases the number of learnable parameters (as we iteratively increase the input size) but the overall architecture still remains independent of the number of objects.

### 4.3 Logic Program Extraction

During evaluation and deployment, both the time and space complexity will increase quickly as the number of constants increases. To speed up inference and have an interpretable model, we post-process the trained model to extract the logical formulas instead of using it directly. For each used module, we replace the

Gumbel-softmax (6) by an argmax to choose the predicates deterministically. The fuzzy operations can then be replaced by their corresponding Boolean ones. Formula extraction can be done recursively from the output of the model. All the non-selected input predicates coming from the previous layer do not need to be computed. A graph containing only the selected predicates is built from the output to the input predicates. The extracted interpretable model can then operate on Boolean tensors, which further saves space and computation time.

## 5 Experimental Results

We performed four series of experiments. The first (resp. second) series evaluate DLM with SOTA baselines on ILP (resp. RL) tasks. The third series correspond to an ablation study that justifies the different components (i.e., critic, Gumbel-softmax, dropout) of our method. The last series present a comparison of the methods in terms of computational costs. Examples of interpretable policies learned by DLM are provided in Appendix B.2. Other details about the experiments (computer specifications, curriculum learning, hyperparameters used in DLM and baselines) are provided in Appendix D.

The first series, which focuses on SL training, confirm that DLM is competitive with SOTA baselines on ILP tasks. Compared to interpretable models (Payani & Fekri, 2019a; Evans & Grefenstette, 2018)), DLM performs better and compared to a SOTA non-interpretable model (Dong et al., 2019), DLM is easier to train successfully. To keep the paper short, we present the details of those results in Appendix B. The results for the other experiments are discussed next.

### 5.1 RL Tasks

For the RL experiments, we justify the baselines we used, provide a short description of the tasks with their corresponding training procedures, explain the performance metrics, and discuss the experimental results.

**RL Baselines**    Our evaluation on ILP tasks suggests that the best neural-logic architectures, apart from DLM, are $\partial$ILP and NLM. While the authors of NLM directly demonstrated it in an RL setting, $\partial$ILP was later extended into an RL method called NLRL (Jiang & Luo, 2019). Based on these observations, we selected NLM and NLRL as strong RL neural-logic baselines to compare DLM with.

**RL Task Description**    Our evaluation is performed on six RL tasks: three Blocks World tasks from Jiang & Luo (2019), three other tasks (two algorithmic tasks and one Blocks World task) from Dong et al. (2019). We provide a short description below, but more details can be found in Appendix C.

In the first three, *Stack*, *Unstack*, and *On*, the agent is trained to learn predicate $Move(X, Y)$, which moves block $X$ on block (or floor) $Y$ if possible. The observable predicates are: $IsFloor(X)$, $Top(X)$, and $On(X, Y)$ with an additional predicate $OnGoal(X, Y)$ for the *On* task only. In *Stack*, the agent needs to stack all the blocks whatever their order. In *Unstack*, the agent needs to put all the blocks on the floor. In *On*, the agent needs to reach the goal specified by *onGoal*.

The last three are *Sorting*, *Path* and *Blocksworld*. In *Sorting*, the agent must learn $Swap(X, Y)$ where $X$ and $Y$ are two elements of a list to sort. The binary observable predicates are *SmallerIndex*, *SameIndex*, *GreaterIndex*, *SmallerValue*, *SameValue*, and *GreaterValue*. In *Path*, the agent is given a graph as a binary predicate with a source node and a target node as two unary predicates. It must learn the shortest path with a unary predicate $GoTo(X)$ where $X$ is the destination node. *Blocksworld* is the most complex task: it features a target world and a source world with numbered blocks, which makes the number of constants to be $2(m + 1)$ where $m$ is the number of blocks and 1 corresponds to the floor. The agent learns $Move(X, Y)$ by moving blocks in the source world. It is rewarded if both worlds match exactly. The binary observable predicates are *SameWorldID*, *SmallerWorldID*, *LargerWorldID*, *SameID*, *SmallerID*, *LargerID*, *Left*, *SameX*, *Right*, *Below*, *SameY*, and *Above*.

All those domains are goal-based sparse-reward RL problems. Since the first three domains are relatively simple, they can be trained and evaluated on fixed instances with a fixed number of blocks. In contrast, for the last three domains, the training and testing instances are generated randomly. Those last three domains,

Table 1: Average rewards of NLRL, NLM, and DLM on RL tasks for the best seed.

| | | Average rewards | | | | | |
|---|---|---|---|---|---|---|---|
| | | NLRL | NLM | nIDLM | DLM | DLM+incr | DLM+incr (SL) |
| Interpretable | | yes | no | no | yes | yes | yes |
| Unstack | 5 vari. | $0.914 \pm 0.01$ | $\mathbf{0.920 \pm 0}$ | $\mathbf{0.920 \pm 0}$ | $\mathbf{0.920 \pm 0}$ | $\mathbf{0.920 \pm 0}$ | $0.920 \pm 0$ |
| Stack | 5 vari. | $0.877 \pm 0.09$ | $\mathbf{0.920 \pm 0}$ | $\mathbf{0.920 \pm 0}$ | $\mathbf{0.920 \pm 0}$ | $\mathbf{0.920 \pm 0}$ | $0.920 \pm 0$ |
| On | 5 vari. | $0.885 \pm 0.01$ | $\mathbf{0.896 \pm 0}$ | $\mathbf{0.896 \pm 0}$ | $\mathbf{0.896 \pm 0}$ | $\mathbf{0.896 \pm 0}$ | $0.896 \pm 0$ |
| Sorting | $m = 10$ | $0.866 \pm 0.1$ | $\mathbf{0.939 \pm 0.02}$ | $\mathbf{0.939 \pm 0.02}$ | $\mathbf{0.939 \pm 0.02}$ | $\mathbf{0.939 \pm 0.02}$ | $0.939 \pm 0.02$ |
| | $M = 50$ | N/A | $\mathbf{0.556 \pm 0.02}$ | $\mathbf{0.556 \pm 0.02}$ | $\mathbf{0.559 \pm 0.02}$ | $\mathbf{0.559 \pm 0.02}$ | $0.559 \pm 0.02$ |
| Path | $m = 10$ | N/A | $\mathbf{0.970 \pm 0}$ | $\mathbf{0.970 \pm 0}$ | $-$ | $\mathbf{0.319 \pm 0.2}$ | $0.970 \pm 0$ |
| | $M = 50$ | | $\mathbf{0.970 \pm 0}$ | $\mathbf{0.970 \pm 0}$ | $-$ | $\mathbf{-0.004 \pm 0.4}$ | $0.970 \pm 0$ |
| Blocksworld | $m = 10$ | N/A | $0.888 \pm 0.02$ | $\mathbf{0.904 \pm 0.02}$ | $-$ | $-$ | $0.894 \pm 0.02$ |
| | $M = 50$ | | $0.153 \pm 0.04$ | $\mathbf{0.159 \pm 0.11}$ | $-$ | $-$ | $0.230 \pm 0.04$ |

N/A: Out of memory issue.     $-$: Could not extract a working interpretable policy for given architecture size.

which are much harder than the first three, require training with curriculum learning (CL), which was also used by Dong et al. (2019). The difficulty of a lesson in CL is defined by the number of objects. Further details about the training with CL are provided in Appendix D.2. After training with $m = 10$, we evaluate the learned model on test instances with $m = 10$, but also $M = 50$ to assess its generalizability.

**RL Metrics**   Like ILP tasks, the performance can be measured in terms of success rates, i.e., percentage of times a trained policy can solve an RL task (i.e., reach a goal). In addition, another natural metric in RL is the average reward obtained during testing.

**RL Results**   Table 1 provides the success rates of all the algorithms on different RL tasks. For DLM, we provide the results obtained by RL training (see Section 4.2.2) where we enforce the convergence to an interpretable policy (DLM) or not (nIDLM). DLM+incr and DLM+incr (SL) correspond to our architecture trained in an incremental way via RL and SL training respectively (see below). Each subrow of an RL task corresponds to some instance(s) on which a trained model is evaluated.

The experimental results show that NLRL does not scale to harder RL tasks, as expected from its performance on ILP tasks. On *Sorting* where we can learn a fully-interpretable policy, DLM is better than NLM in terms of computational time and memory usage during testing. Thus, DLM is always superior to NLRL and can match the performance of NLM for simpler tasks.

For the harder RL tasks (*Path*, *Blocksworld*), when we do not enforce the convergence to an interpretable policy, our method with RL training (nIDLM) can reach similar or better performances than NLM. Interestingly, although both NLM and nIDLM learn a non-interpretable policy, nIDLM can generalize better than NLM in *Blocksworld*, which suggests that the architectural inductive bias of DLM is more suitable for problems described with FOL. However, obtaining a good interpretable policy with CL reveals to be difficult: there is a contradiction between learning to solve a lesson and converging to a final interpretable policy that generalizes. Indeed, on the one hand, we can learn an interpretable policy for a lesson with a small number of objects, however that policy will probably not generalize and training that interpretable policy on the next lesson is hard since the softmaxes are nearly argmaxes. On the other hand, we can learn to solve all the lessons with a non-interpretable policy, but that final policy is hard to turn into an interpretable one, because of the many local optima in the loss landscape. This training difficulty explains why we did not manage to learn an interpretable policy for *Path* and *Blocksworld* using RL training.

To demonstrate that this issue is due to RL training (and not to our DLM architecture), we also evaluated it with incremental training (see Section 4.2.3). In the incremental training phases, we tried both SL and

Table 2: Average percentage of successful seeds (PSS) and average success rates (SR) on all the tasks of Family Tree and Graph Reasoning during testing with interpretable rules. Score computed with 5 seeds for each task.

|  | Average PSS (%) | Average SR (%) |
|---|---|---|
| Softmax without noise | 58 | $97.3 \pm 6.1$ |
| Constant $\beta$ with Dropout | 68 | $97.7 \pm 9.3$ |
| Without Dropout | 70 | $99.3 \pm 1.6$ |
| Gaussian noise | 80 | $99.7 \pm 0.8$ |
| DLM | **95** | **99.8** $\pm$ **0.7** |

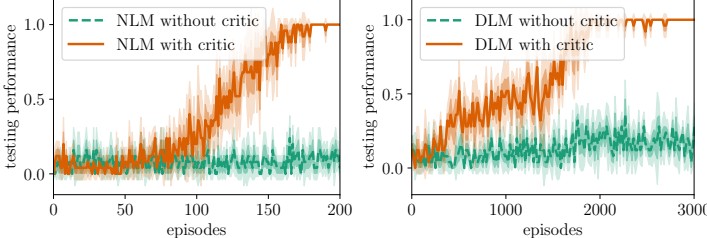

Figure 3: Learning performance with or without our proposed critic with NLM (left) and DLM (right).

RL training. While incremental RL training does increase the performance compared to RL training alone, notably in *Path*, it is not sufficient to solve *Blocksworld* nor *Path* completely.

However, incremental SL training is successful, as reported in column DLM+incr (SL) in Table 1. It corresponds to an imitation learning scenario where the agent tries to copy a good policy given by a teacher (possibly the one learned by nIDLM which does not enforce interpretability). With DLM+incr (SL), unlike vanilla DLM and DLM+incr, we were able to extract a good interpretable policy for *Blocksworld*. It needed a stacking of 4 DLMs of depth 8, which shows the difficulty of finding an interpretable formula for this task. Interestingly, the generalization performance of DLM+incr (SL) is the best on *Blocksworld*, which suggests that enforcing interpretability is beneficial. We leave for future work the investigation of alternative RL training methods that scale better than CL for sparse-reward RL problems like *Path* and *Blocksworld*.

## 5.2 Ablation and Computational Cost Studies

For simplicity, these studies are performed in ILP tasks. Their description can be found on Appendix B.

**Ablation** We performed an ablation study to understand the different features (e.g., Gumbel noise in softmaxes, decreasing $\beta$, use of Dropout) in DLM. We trained our model on ILP tasks by using only softmax without injecting noise, without decreasing the noise over time, without having a dropout noise and finally by replacing the Gumbel distribution with a Gaussian one. In those experiments, during evaluation, we still used an argmax to retrieve the interpretable rules. Table 2 shows that all our choices help our model reach interpretable rules. Success rate (SR) is the proportion of examples well-classified by a trained model. Percentage of successful seeds (PSS) is the proportion of seeds for which a trained model reaches an SR of 100%.

In addition, to evaluate the quality of our proposed critic architecture, we used it to train both NLM and DLM. Fig. 3 shows the testing performance of NLM or DLM trained with REINFORCE or with an actor-critic scheme with our proposed critic architecture. Recall the testing performance corresponds to the evaluation of the latest trained policy in terms of success rates at regular training steps. It has been averaged over 3 environments and 5 random seeds: *Unstack*, *Stack*, and *On*. As shown in Fig. 3, using our proposed critic architecture greatly accelerates the RL training for both NLM and DLM.

**Comparative computational costs** We compare now the different algorithms with respect to computational times and memory usage during testing (Figure 4) and training (see Table 5 in Appendix). During

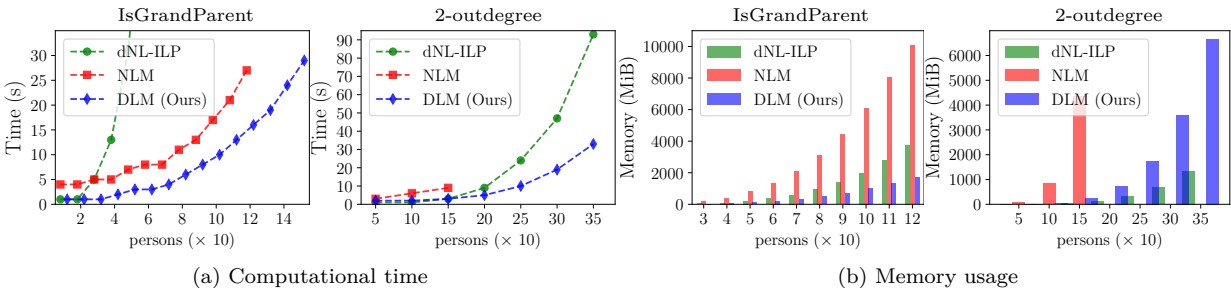

Figure 4: Comparison during test in *IsGrandParent* and *2-outdegree*. On *2-outdegree*, NLM is rapidly out of memory and **dNL-ILP does not achieve a 100% success rate**.

testing, we extracted the compact representation of the produced solution by DLM (see Section 4.3). Figure 4 clearly shows that DLM scales better than the other baselines in terms of both memory and computational times. Note that in the second task (*2-outdegree*), dNL-ILP does not achieve a 100% success rate and searches instead in a much smaller space.

## 6   Conclusion

We proposed a novel neural-logic architecture that is capable of learning a fully-interpretable solution, i.e., logic program. Among differentiable methods, it obtains state-of-the-art results for inductive logic programming tasks, while retaining interpretability and scaling much better. For reinforcement learning tasks, it is superior to previous interpretable neuro-logic models. Compared to non-interpretable models, it can achieve comparable or higher performances using incremental training. Moreover, it can generalize better, and more importantly, it scales much more advantageously in terms of computational times and memory usage during testing.

Our results suggest that learning a fully-interpretable solution using supervised learning with our approach could be practical, but enforcing interpretability in more complex reinforcement learning tasks, especially with RL training using sparse rewards, is a much harder problem, which calls for novel neural-logic architectures or novel training techniques. Another interesting phenomenon that we have observed is that there is a tension between learning interpretable solutions and curriculum learning. We leave a study of this phenomenon for future work.

### Acknowledgments

This work has been supported in part by a project funded by Huawei (HF2020055001) and the program of National Natural Science Foundation of China (No. 62176154).

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

# A    Details About DLM

## A.1    Tensor Representation

DLM processes predicates by working on the concatenation of their corresponding tensors. We define below the concatenation, provide some illustrative examples, and list the pseudo-code (Algorithm 1) for the inference in DLM.

A set $\mathcal{P}_b$ of $b$-ary predicates can be represented as a tensor, denoted $\boldsymbol{\mathcal{P}}_b$, of order $b + 1$ with shape $[m, \ldots, m, |\mathcal{P}_b|]$.

The concatenation of two tensors $\boldsymbol{\mathcal{P}}_b$ and $\boldsymbol{\mathcal{Q}}_b$ representing two sets of $b$-ary predicates, $\mathcal{P}_b$ and $\mathcal{Q}_b$, is performed over the last dimension. It results in a tensor of order $b + 1$ with shape $[m, ...m, |\mathcal{P}_b| + |\mathcal{Q}_b|]$.

We provide some examples of how the different operations (i.e., expansion, reduction, permutation) work on tensors:

**Example 4.** ***Expansion:*** *A unary predicate $Top(X)$ represented over 3 constants with the vector $(0, 0, 1)$ would be expanded into $\widehat{Top}(X, Y)$ as the following matrix $((0, 0, 0), (0, 0, 0), (1, 1, 1))$.*

***Reduction:*** *A binary predicate $On(X, Y)$ represented with the matrix $((0, 1, 0), (0, 0, 1), (0, 0, 0))$ would be reduced to the vector $(1, 1, 0)$ with the existential quantifier representing whether $X$ is "On" any objects.*

***Permutation:*** *A binary predicate $On(X, Y)$ represented with the matrix $((0, 1, 0), (0, 0, 1), (0, 0, 0))$ would be permuted to the matrix $((0, 0, 0), (1, 0, 0), (0, 1, 0))$ representing the relation $Under(X, Y)$.*

We provide a detailed example for understanding DLM's architecture:

**Example 5.** *Consider a blocks world environment with $m = 4$ objects: 3 blocks $(u, v, w)$ and the floor $(floor)$. Assuming that only two initial predicates are available, the following facts $\{On(u, v), On(v, floor), On(w, floor), Top(u), Top(w)\}$ are encoded by two tensors. The first tensor encoding the unary predicate $Top(X)$ is a vector of length $4$, since there are $4$ objects. The second tensor for the binary predicate $On(X, Y)$ is a $4 \times 4$-matrix. Those two tensors will feed the DLM network at layer 0 on different breadths (1 and 2 respectively).*

*We first focus on what happens in the first layer of breadth 1 $(l = 1, b = 1)$. By assumption, there is no expansion. Since the previous layer $l-1$ with breadth $b+1$ is not empty, the reduction operation generates two predicates $OnR1(X) \leftarrow \forall Y, On(X, Y)$ and $OnR2(X) \leftarrow \exists Y, On(X, Y)$. Hence, the possible (positive) input predicates $\mathcal{R}_1^1$ are $\{Top(X), OnR1(X), OnR2(X)\}$. Since we deal with unary predicates, the permutation operation does not play a role.*

*Now, consider the first layer of breadth 2 $(l = 1, b = 2)$. By assumption, there is no reduction. The expansion operation generates one predicate $TopE(X, Y)$ whose truth value is given by $Top(X)$. Therefore, after the permutation operation, $\mathcal{R}_2^1 = \{On(X, Y), TopE(X, Y), On(Y, X), TopE(Y, X)\}$.*

*Assume that we want to detect the objects that are neither the floor, nor on the top of a stack (i.e., $v$ here) with a unary predicate, which we call $BlockNotTop(X)$. This predicate can be invented as a conjunction with a negative first atom. As $|\mathcal{R}_1^1| = 3$, $w_P$ and $w_{P'}$ are vectors of length $4$ due to the preservation operation. To build $BlockNotTop(X) \leftarrow \neg Top(X) \wedge OnR2(X)$, $w_P$ should be zero everywhere except for selecting $\neg Top(X)$ in $\{\mathbb{T}\} \cup \eta(R_b^l)$. Accordingly, $w_{P'}$ should be zero everywhere except for selecting $OnR2(X)$ inside $\{\mathbb{T}\} \cup \mathcal{R}_b^l$.*

We provide the formulas for the inference of NLM and DLM units in the first layer associated with Example 5 in Table 3.

## A.2    Interpretability and Expressivity

**Interpretation**    Following NLM, we keep the probabilistic interpretation of the tensor representations of the predicates in DLM. This interpretation justifies the application of statistical machine learning techniques to train DLM via cross-entropy minimization for supervised tasks (i.e., ILP) for instance. In this interpretation, using the fuzzy conjunctions and disjunctions can be understood as making the assumption

Table 3: First layer of NLM and DLM on Example 5. The input predicates are composed of Top(X), represented by a vector $U_1$ of length $m$, and On(X,Y) represented a $m \times m$ matrix $U_2$. The operation $||$ denotes the concatenation of tensors on a new dimension. Note that $min$ and $max$ operations for the reduction are always performed on the last dimension of the input tensor. The expansion operation is denoted $\zeta$, it repeats a tensor $m$ times on a new last dimension. The permutation operation is denoted $\nu$ and permutes a tensor according to all the possible permutations without repetitions stacked in a new dimension. For instance, $\zeta(U_2)$, a $m \times m \times m$ tensor, would result in a new $m \times m \times m \times 3!$ tensor. Note that for DLM we assumed $n_A = 2$ for readability.

| Output breadth | NLM | DLM |
|---|---|---|
| 0 (nullary predicates) | $\mathrm{MLP}(min(U_1)||max(U_1))$ | $\mathrm{FuzzyAND}_{\theta,\tau}(U_3, U_3)$ $\mathrm{FuzzyOR}_{\theta,\tau}(U_3, U_3)$ with $U_3 = min(U_1)||max(U_1)||\mathbb{1}$ |
| 1 (unary predicates) | $\mathrm{MLP}(min(U_2)||max(U_2)||U_1)$ | $\mathrm{FuzzyAND}_{\theta,\tau}(U_4, U_4)$ $\mathrm{FuzzyAND}_{\theta,\tau}(U_4, 1 - U_4)$ $\mathrm{FuzzyOR}_{\theta,\tau}(U_4, U_4)$ $\mathrm{FuzzyOR}_{\theta,\tau}(U_4, 1 - U_4)$ with $U_4 = min(U_2)||max(U_2)||U_1||\mathbb{1}$ |
| 2 (binary predicates) | $\mathrm{MLP}(U_2||U_2^T||\zeta(U_1))$ | $\mathrm{FuzzyAND}_{\theta,\tau}(U_5, U_5)$ $\mathrm{FuzzyAND}_{\theta,\tau}(U_5, 1 - U_5)$ $\mathrm{FuzzyOR}_{\theta,\tau}(U_5, U_5)$ $\mathrm{FuzzyOR}_{\theta,\tau}(U_5, 1 - U_5)$ with $U_5 = U_2||U_2^T||\zeta(U_1)||\mathbb{1}$ |
| 3 (ternary predicates) | $\mathrm{MLP}(\nu(\zeta(U_2)))$ | $\mathrm{FuzzyAND}_{\theta,\tau}(U_6, U_6)$ $\mathrm{FuzzyAND}_{\theta,\tau}(U_6, 1 - U_6)$ $\mathrm{FuzzyOR}_{\theta,\tau}(U_6, U_6)$ $\mathrm{FuzzyOR}_{\theta,\tau}(U_6, 1 - U_6)$ with $U_6 = \nu(\zeta(U_2))||\mathbb{1}$ |

---

**Algorithm 1** Inference of a module at layer $l$ and breadth $b$

---

**Require:** $\mathcal{P}_{b-1}^{l-1}$, $\mathcal{P}_b^{l-1}$, $\mathcal{P}_{b+1}^{l-1}$ (tensor representations of the sets of predicates at layer $l-1$ with breadth $b-1$, $b$, and $b+1$ respectively), $\theta$ (parameters of the module) and $\tau$ (temperature)
**Ensure:** $\mathcal{P}_b^l$ (tensor representation of the set of the output predicates)

$\quad \check{\mathcal{P}}_{b+1}^{l-1} \leftarrow \mathrm{Reduction}(\mathcal{P}_{b+1}^{l-1}).$
$\quad \hat{\mathcal{P}}_{b-1}^{l-1} \leftarrow \mathrm{Expansion}(\mathcal{P}_{b-1}^{l-1}).$
$\quad \mathcal{Q}_b^l \leftarrow \mathrm{Permutation}(\mathrm{Concatenation}(\check{\mathcal{P}}_{b+1}^{l-1}, \mathcal{P}_b^{l-1}, \hat{\mathcal{P}}_{b-1}^{l-1}))$
$\quad \mathcal{Q}_b^l \leftarrow \mathrm{Preservation}(\mathcal{Q}_b^l)$
$\quad \mathcal{R}_b^l \leftarrow \mathrm{Negation}(\mathcal{Q}_b^l)$
$\quad$ // Assuming $n_A = 2$ for readability
$\quad \mathcal{F}_1 \leftarrow \mathrm{FuzzyAND}_{\theta,\tau}(\mathcal{Q}_b^l, \mathcal{Q}_b^l)$ $\qquad\qquad$ Eq. (3)
$\quad \mathcal{F}_2 \leftarrow \mathrm{FuzzyAND}_{\theta,\tau}(\mathcal{Q}_b^l, \mathcal{R}_b^l)$ $\qquad\qquad$ Eq. (3)
$\quad \mathcal{F}_3 \leftarrow \mathrm{FuzzyOR}_{\theta,\tau}(\mathcal{Q}_b^l, \mathcal{Q}_b^l)$ $\qquad\qquad$ Eq. (4)
$\quad \mathcal{F}_4 \leftarrow \mathrm{FuzzyOR}_{\theta,\tau}(\mathcal{Q}_b^l, \mathcal{R}_b^l)$ $\qquad\qquad$ Eq. (4)
$\quad \mathcal{P}_b^l \leftarrow \mathrm{Concatenation}(\mathcal{F}_1, \mathcal{F}_2, \mathcal{F}_3, \mathcal{F}_4)$
$\quad$ **return** $\mathcal{P}_b^l$

---

of probabilistic independence between the truth values of any pairs of atoms in $\mathcal{R}_b^l$. This may seem a strong assumption, however this is not detrimental since we want to learn a logic program operating on Boolean tensors.

**Expressivity**  Previous work like $\partial$ILP or NLM (Evans & Grefenstette, 2018; Jiang & Luo, 2019; Dong et al., 2019) can express Datalog programs, a subset of FOL composed of Horn clauses that do not contain function symbols. With the negation operation, DLM is more expressive than $\partial$ILP. With sufficient breadth $B$ and depth $L$, it can express any normal logic programs, i.e., Horn clauses with negative literals (Kowalski, 2014), that do not contain function symbols. We refer the reader to the proof present in (Dong et al., 2019), its extension with the negation is trivial. Hence, the only FOL formulas not covered by DLM are the ones containing function symbols.

In practice, the expressivity of DLM can not only be controlled by setting hyperparameters $L$, $B$, $n_A$, and $n_O$ but also by restricting the inputs of logic modules (e.g., no negation or no existential or universal quantifiers). Therefore, a priori knowledge can be injected in this architecture by choosing different values for $B$ at each layer, different values for $n_O$ for each computation unit, different values for $n_A$ for each logic module, or by removing some inputs of logic modules for instance.

### A.3  Implementation Remarks

In our implementation, half of the $n_O$ outputs of each computation unit corresponds to *and* logic modules and the other half corresponds to *or* modules. Moreover, to enforce a stronger bias, among the *and* (resp. *or*) logic modules, half of them only takes $\mathcal{Q}_b^l \cup \{\mathbb{T}\}$ (resp. $\mathcal{Q}_b^l \cup \{\mathbb{F}\}$) as inputs. Note that there is no expressivity loss with this bias, as long as the architecture is large enough.

For the other half of the *and* (resp. *or*) modules, half of the $n_A$ terms used to build a conjunction (resp. disjunction) is from $\mathcal{Q}_b^l \cup \{\mathbb{T}\}$ (resp. $\mathcal{Q}_b^l \cup \{\mathbb{F}\}$) and the other half is from $\eta(\mathcal{Q}_b^l) \cup \{\mathbb{T}\}$ (resp. $\eta(\mathcal{Q}_b^l) \cup \{\mathbb{F}\}$).

### A.4  Training Algorithms

For completeness, we provide the pseudo-code of the training algorithms we used. They are mostly standard, apart from the incremental training technique.

Here is the pseudo-code for supervised training:

---

**Algorithm 2** Supervised training of DLM

---

**Require:** $(\mathcal{P}^0, P^T)$ (input and target predicates describing an ILP task), $\boldsymbol{\theta}$ (randomly-initialized DLM parameters), with hyperparameters $\tau_i$ (softmax temperature), $\beta_i$ (Gumbel scale), $p_i$ (dropout probability), $\alpha_i$ (learning rate)
**Ensure:** $\boldsymbol{\theta}$ (trained DLM parameters)
   **for** $i = 0, 1, 2, \ldots$ **do**
      Sample a batch of ILP instances described by atoms $(\boldsymbol{\mathcal{P}}^0, \boldsymbol{P}^T)$
      $\hat{\boldsymbol{P}}^T \leftarrow \mathrm{DLM}_{\boldsymbol{\theta}}(\boldsymbol{\mathcal{P}}^0)$ with softmax temperature $\tau_i$, Gumbel noise $\beta_i$ and dropout probability $p_i$
      Compute binary cross-entropy loss $\mathcal{L}_{\boldsymbol{\theta}}(\boldsymbol{P}^T, \hat{\boldsymbol{P}}^T)$
      $\boldsymbol{\theta} \leftarrow \boldsymbol{\theta} - \alpha_i \nabla_{\boldsymbol{\theta}} \mathcal{L}_{\boldsymbol{\theta}}(\boldsymbol{P}^T, \hat{\boldsymbol{P}}^T)$
   **end for**
   **return** $\boldsymbol{\theta}$

---

The algorithm could naturally also be applied to train on a fixed unique ILP instance. The hyperparameters are scheduled to decrease so that a convergence to a fully interpretable solution is possible (see Table 10 for more details).

Here is the pseudo-code for RL training, which is actually based on PPO (Schulman et al., 2017):

---

**Algorithm 3** RL training of DLM

---

**Require:** $(\mathcal{P}^0, P^A)$ (predicates describing an RL task), $\boldsymbol{\theta}$ (randomly-initialized DLM parameters), $\boldsymbol{\psi}$ (randomly-initialized critic parameters), with hyperparameters $\tau_i$ (softmax temperature), $\beta$ (Gumbel scale), $p$ (dropout probability), $\alpha$ and $\rho$ (learning rates)
**Ensure:** $\boldsymbol{\theta}$ (trained DLM parameters)
  **for** $i = 0, 1, 2, \ldots$ **do**
    **for** $j = 1, 2, \ldots, N$ **do**
      Sample initial state $s_0^j$ (described by $\mathcal{P}^0$)
      Generate $j$-th trajectory $(s_0^j, s_1^j, \ldots)$ using DLM$_{\boldsymbol{\theta}}$ with softmax temperature $\tau_i$, Gumbel noise $\beta_i$ and dropout probability $p_i$
      Compute rewards-to-go $R_t^j$ of $j$-th trajectory (i.e., 1 if goal reached and 0 otherwise)
    **end for**
    Compute $\hat{A}$ from $(R_t^j)_{j,t}$ and critic $(v^{\boldsymbol{\psi}}(s_t^j))_{j,t}$ with GAE
    $\boldsymbol{\theta} \leftarrow \boldsymbol{\theta} + \alpha \nabla_{\boldsymbol{\theta}} J_{PPO}(\theta)$
    $\boldsymbol{\psi} \leftarrow \boldsymbol{\psi} - \rho \nabla_{\boldsymbol{\psi}} MSE_\psi(\hat{A} + v(s_t^j)_{j,t}, v^{\boldsymbol{\psi}}(s_t^j)_{j,t})$
  **end for**
  **return** $\boldsymbol{\theta}$

---

Recall in PPO, the update of the parameters of the critic is performed by minimizing the mean square error (MSE) between the output of the critic and the Generalized Advantage Estimate (GAE) (Schulman et al., 2017) through gradient descent.

Here is the pseudo-code for incremental training:

---

**Algorithm 4** Incremental training of DLM

---

**Require:** $(\mathcal{P}^0, P^A)$ (predicates describing an RL task)
**Ensure:** $\bar{\mathcal{P}}^0$ (initial predicates and those invented during incremental training), $\boldsymbol{\theta}$ (trained DLM parameters)

  $\bar{\mathcal{P}}^0 = \mathcal{P}^0$
  **for** $k = 0, 1, 2, \ldots$ **do**
    Randomly initialize $\boldsymbol{\theta}$
    $\boldsymbol{\theta} \leftarrow$ train DLM$_{\boldsymbol{\theta}}$ on $(\mathcal{P}^0, P^A)$
    **if** performance is satisfying **then**
      **return** $(\bar{\mathcal{P}}^0, \boldsymbol{\theta})$
    **else**
      $\mathcal{P}^{(k)} \leftarrow$ extract predicates used in DLM$_{\boldsymbol{\theta}}$
      $\bar{\mathcal{P}}^0 = \bar{\mathcal{P}}^0 \cup \mathcal{P}^{(k)}$
    **end if**
  **end for**

---

Incremental training is written here for solving an RL task, but it could be adapted to solve a hard ILP task as well. Inside the for loop, DLM can be trained via SL or RL, depending on the training method the necessary hyperparameters need to be passed. In addition for RL training, the critic parameters would also be needed and should be randomly initialized before RL training.

# B   Additional Experimental Results and Details

## B.1  ILP Tasks

For the ILP experiments, we justify the baselines we selected, provide a short description of the tasks, explain the performance metrics used for the evaluation, and discuss the results we obtained in terms of performance and computational costs. During this discussion, we provide the training details.

**ILP Baselines** Since DLM is a neural-logic architecture, we only compare with other neural-logic approaches: $\partial$ILP (Evans & Grefenstette, 2018), NLM (Dong et al., 2019), and dNL-ILP (Payani & Fekri, 2019a). Differentiable architectures such as MEM-NN (Sukhbaatar et al., 2015) or DNC (Graves et al., 2016) are not included, since they have been shown to be inferior on ILP tasks compared to NLM and they, furthermore, do not provide any interpretable solutions. Also, the approaches in multi-hop reasoning (Yang et al., 2017; Yang & Song, 2020) are also left out because although they can scale well, the rules they can learn are much less expressive, which prevent them from solving any complex ILP tasks in an interpretable way. Besides, DeepProbLog (Manhaeve et al., 2018), which is based on backward chaining, is not included since it is not easy to perform predicate invention with it.

**ILP Task Description** For the ILP tasks, we evaluate on two domains: Family Tree and Graph Reasoning. In the Family Tree domain, different tasks are considered corresponding to different target predicates to be learned from an input graph where nodes representing individuals are connected with relations: *IsMother*$(X,Y)$, *IsFather*$(X,Y)$, *IsSon*$(X,Y)$, and *IsDaughter*$(X,Y)$. The target predicates are *HasFather*, *HasSister*, *IsGrandParent*, *IsUncle*, and *IsMGUncle* (i.e., maternal great uncle). In the Graph Reasoning domain, the different target predicates to be learned from an input graph are *AdjacentToRed*, *4-Connectivity*, *6-Connectivity*, *1-OutDegree*, *2-OutDegree* (see Appendix C.1 for their definitions).

**ILP Metrics** The performance of an ILP method on a task can be measured in terms of success rate, which is the percentage of relations in a test instance that are correctly classified by a trained model. Following previous work, we report it as an average over 250 random instances for the best model obtained over 10 random seeds. In addition, we also report the percentage of successful seeds (PSS), which is the percentage of those 10 seeds that reach a 100% success rate on the testing instances. PSS indicates how reliable a model and its training are.

**Performance on ILP Tasks** In Table 4, we report those two metrics for DLM and the baselines we selected. All the methods are trained on random instances with the number of constants $m = 20$, except for $\partial$ILP. Since its authors did not release its source code, the reported success rates for $\partial$ILP are from Dong et al. (2019) and its PSS is missing. For NLM and dNL-ILP, we use the source codes shared by their authors. For the latter, Payani & Fekri (2019a) did not evaluate their method in any standard ILP tasks. Using their source code, we did our best to find the best set of hyperparameters (see Appendix D.3) for each ILP task.

Column $m = 20$ provides the success rates for test instances with the same number of constants as in the training instances, while column $M = 100$ provides the success rates for test instances with 100 constants, which allow to measure the generalization capability of the trained models. N/A means that the method ran out of memory. For dNL-ILP, the memory issue comes from the fact that, both learning auxiliary predicates and increasing the number of variables of predicates increase memory consumption sharply with a growing number of constants (see details in Appendix D.3.1).

The experimental results demonstrate that the previous interpretable methods, dNL-ILP and $\partial$ILP, do not scale to difficult ILP tasks and to a larger number of constants. However, DLM can solve all the ILP tasks like NLM, while DLM can in addition provide an interpretable rule in contrast to NLM. Moreover, we can observe that DLM is more stable, in terms of successful seeds, than all the other methods including NLM. To further demonstrate the stability of DLM, we also report the standard deviations of its success rates over the different seeds in Table 4 (values in parentheses). They show that even when a 100% success rate is not reached for a given seed, the trained model still reaches a success rate close to 100%.

**Computational Costs on ILP Tasks** In Table 5, we provide the computational time (column T) and memory usage (column M) during training and testing on two different representative ILP tasks: *IsGrandparent* and *2-Outdegree*. The Training rows provides the computational costs observed during training while the subsequent rows give the costs measured during testing for different numbers of constants. The dNL-ILP method can scale very well, but both NLM and DLM perform much better than dNL-ILP as shown in our previous experiments. While the training costs for DLM are slightly higher than NLM, they are still reasonable. More interestingly, DLM scales much better than NLM during testing, which is the most important, since these computational costs are incurred after the trained model is deployed.

Table 4: Success rates (%) and percentage of successful seeds of dNL-ILP, $\partial$ILP, NLM, and DLM on Family Tree and Graph Reasoning.

| | dNL-ILP | | | $\partial$ILP | | NLM | | | DLM (Ours) | | |
|---|---|---|---|---|---|---|---|---|---|---|---|
| Family Tree | $m=20$ | $M=100$ | PSS | $m=20$ | $M=100$ | $m=20$ | $M=100$ | PSS | $m=20$ | $M=100$ | PSS |
| *HasFather* | 100 | 100 | 100 | 100 | 100 | 100 | 100 | 100 | 100 ($\pm0$) | 100 ($\pm0$) | 100 |
| *HasSister* | 100 | 100 | 40 | 100 | 100 | 100 | 100 | 100 | 100 ($\pm0$) | 100 ($\pm0$) | 100 |
| *IsGrandparent* | 100 | 100 | 80 | 100 | 100 | 100 | 100 | 100 | 100 ($\pm0$) | 100 ($\pm0$) | 100 |
| *IsUncle* | 97.32 | 96.77 | 0 | 100 | 100 | 100 | 100 | 90 | 100 ($\pm0$) | 100 ($\pm0$) | 100 |
| *IsMGUncle* | 99.09 | N/A | 0 | 100 | 100 | 100 | 100 | 20 | 100 ($\pm0.01$) | 100 ($\pm0.08$) | 70 |
| Graph Reasoning | $m=10$ | $M=50$ | PSS | $m=10$ | $M=50$ | $m=10$ | $M=50$ | PSS | $m=10$ | $M=50$ | PSS |
| *AdjacentToRed* | 100 | 100 | 100 | 100 | 100 | 100 | 100 | 90 | 100 ($\pm0.02$) | 100 ($\pm0.01$) | 90 |
| *4-Connectivity* | 91.36 | 85.30 | 0 | 100 | 100 | 100 | 100 | 100 | 100 ($\pm0$) | 100 ($\pm0$) | 100 |
| *6-Connectivity* | 92.80 | N/A | 0 | 100 | 100 | 100 | 100 | 60 | 100 ($\pm0.00$) | 100 ($\pm0$) | 90 |
| *1-OutDegree* | 82.00 | 78.44 | 0 | 100 | 100 | 100 | 100 | 100 | 100 ($\pm0$) | 100 ($\pm0$) | 100 |
| *2-OutDegree* | 83.39 | 8.24 | 0 | N/A | N/A | 100 | 100 | 100 | 100 ($\pm0$) | 100 ($\pm0$) | 100 |

N/A: Out of memory issues.     PSS: Percentage of Successful Seeds reaching 100% of success rates on the testing instances.

Table 5: Computational costs of dNL-ILP, NLM, and DLM on *IsGrandParent* and *2-Outdegree*.

| | dNL-ILP | | NLM | | DLM (Ours) | |
|---|---|---|---|---|---|---|
| *IsGrandparent* | T | M | T | M | T | M |
| Training | 201 | 30 | 1357 | 70 | 1629 | 382 |
| $m=10$ | 1 | 27 | 4 | 24 | 1 | 2 |
| $m=20$ | 1 | 30 | 4 | 70 | 1 | 24 |
| $m=30$ | 5 | 42 | 5 | 188 | 1 | 24 |
| $m=40$ | 13 | 99 | 5 | 414 | 2 | 74 |
| $m=50$ | 31 | 198 | 7 | 820 | 3 | 124 |
| $m=60$ | 65 | 358 | 8 | 1341 | 3 | 226 |
| $m=70$ | 119 | 596 | 8 | 2089 | 4 | 344 |
| $m=80$ | 192 | 932 | 11 | 3123 | 6 | 500 |
| $m=90$ | 303 | 1390 | 13 | 4434 | 8 | 724 |
| $m=100$ | 464 | 1994 | 17 | 6093 | 10 | 1002 |
| $m=110$ | 656 | 2771 | 21 | 8079 | 13 | 1321 |
| $m=120$ | 915 | 3751 | 27 | 10056 | 16 | 1710 |
| $m=130$ | 1247 | 4964 | N/A | N/A | 19 | 2161 |
| *2-Outdegree* | T | M | T | M | T | M |
| Training | 966 | 22 | 2522 | 844 | 3238 | 1372 |
| $m=5$ | 1 | 22 | 3 | 78 | 2 | 4 |
| $m=10$ | 1 | 22 | 6 | 844 | 2 | 40 |
| $m=15$ | 3 | 42 | 9 | 4342 | 3 | 254 |
| $m=20$ | 9 | 112 | N/A | N/A | 5 | 732 |
| $m=25$ | 24 | 314 | N/A | N/A | 10 | 1751 |
| $m=30$ | 47 | 682 | N/A | N/A | 19 | 3594 |
| $m=35$ | 93 | 1332 | N/A | N/A | 33 | 6666 |

T: time (s), M: Memory (MB).     DLM used depth 4, breadth 3 for *IsGrandparent*, and depth 6, breadth 4 for *2-Outdegree*.     N/A: Out of memory issues.

### B.2 Examples of Interpretable Rules or Policies

#### B.2.1 Discussion of Two Examples

As illustration for ILP, we provide the logic program learned by our method on the task *IsGrandParent*. We used $L = 5$ layers, $B = 3$ breadth, $n_A = 2$ atoms, and $n_O = 8$ outputs per logic modules. For better legibility, we give more meaningful names to the learned rules and remove the expansions and reductions:

$$IsChild1(a,b) \leftarrow IsSon(a,b) \vee IsDaughter(a,b)$$
$$IsChild2(a,b) \leftarrow IsSon(a,b) \vee IsDaughter(a,b)$$
$$IsGCP(a,b,c) \leftarrow IsChild2(a,c) \wedge IsChild2(c,b)$$
$$IsGPC1(a,b,c) \leftarrow IsChild1(c,a) \wedge IsChild(b,c)$$
$$IsGPC2(a,b,c) \leftarrow IsGPC1(a,b,c) \vee IsGCP(b,a,c)$$
$$IsGP(a,b) \leftarrow \exists C, IsGPC2(a,b,C) \wedge \exists C, IsGPC2(a,b,C)$$
$$IsGrandParent(a,b) \leftarrow IsGP(a,b) \wedge IsGP(a,b)$$

The logic program extracted from the trained DLM has redundant parts (e.g., $P \wedge P$), because we used a relatively large architecture to ensure sufficient expressivity. Note that being able to learn interpretable solutions with a large architecture is a desirable feature when the designer does not know the solution beforehand. Redundancy could be reduced by using a smaller architecture, otherwise the redundant parts (e.g., $P \wedge P$) could easily be removed by post-processing the extracted logic program, as we did. After simplification, the solution is given by the following program, which shows that the target predicate has been perfectly learned:

$$IsChild(a,b) \leftarrow IsSon(a,b) \vee IsDaughter(a,b)$$
$$IsGPC1(a,b,c) \leftarrow IsChild(c,a) \wedge IsChild(b,c)$$
$$IsGrandParent(a,b) \leftarrow \exists C, IsGPC1(a,b,C)$$

As an illustration for RL, we provide the simplified logic program learned by our method on the task *On*, which corresponds to the output of the reasoning part:

$$Move(a,b) \leftarrow (OnGoal(a,b) \vee IsFloor(b)) \wedge$$
$$\neg On(a,b) \wedge Top(a).$$

Using this program, the decision-making part (stochastically) moves blocks to the floor and moves the good block on its goal position when it can. For completeness, we provide the complete logic program:

$$Pred1(a,b) \leftarrow OnGoal(b,a) \vee IsFloor(a)$$
$$Pred2(a,b) \leftarrow \neg On(a,b) \wedge Top(a)$$
$$Pred3(a,b) \leftarrow Pred1(b,a) \wedge Pred2(a,b)$$
$$Pred4(a,b) \leftarrow Pred1(b,a) \wedge Pred1(b,a)$$
$$Move(a,b) \leftarrow Pred3(a,b) \wedge Pred4(a,b)$$

Being able to find solutions in a large architecture is a desirable feature when the designer does not know the solution before hand. Besides, note that we directly output an interpretable logic program. In contrast, with previous interpretable models, logic rules with high weights are extracted to be inspected. However, those rules may not generalize because weights are usually not concentrated on one element.

### B.2.2 Other ILP Examples

Here are other examples on the family tree domain:

$$Pred1(a) \leftarrow \exists B, IsFather(a, B) \land \exists B, IsMother(a, B)$$
$$Pred2(a) \leftarrow \exists B, IsFather(a, B) \lor \exists B, IsMother(a, B)$$
$$Pred3(a) \leftarrow \exists B, IsMother(a, B) \lor \exists B, IsMother(a, B)$$
$$Pred4(a) \leftarrow Pred1(a) \lor Pred2(a)$$
$$Pred5(a) \leftarrow Pred3(a) \lor Pred3(a)$$
$$Pred6(a) \leftarrow Pred4(a) \land Pred5(a)$$
$$Pred7(a) \leftarrow Pred6(a) \lor Pred6(a)$$
$$Pred8(a) \leftarrow Pred6(a) \lor Pred6(a)$$
$$HasFather(a) \leftarrow Pred7(a) \land Pred8(a)$$

$$Pred1(a, b) \leftarrow IsDaughter(b, a) \land IsMother(a, b)$$
$$Pred2(a, b) \leftarrow IsDaughter(b, a) \land IsFather(a, b)$$
$$Pred3(a, b) \leftarrow IsDaughter(b, a) \lor IsMother(a, b)$$
$$Pred4(a, b) \leftarrow \exists C, Pred2(b, a, C) \land \exists C, Pred1(b, a, C)$$
$$Pred5(a, b) \leftarrow \exists C, Pred3(b, a, C) \land \exists C, Pred1(b, a, C)$$
$$Pred6(a) \leftarrow \exists B, Pred4(a, B) \land \exists B, Pred5(a, B)$$
$$Pred7(a) \leftarrow Pred6(a) \lor Pred6(a)$$
$$HasSister(a) \leftarrow Pred7(a) \land Pred7(a)$$

$$Pred1(a, b) \leftarrow IsSon(b, a) \land IsSon(b, a)$$
$$Pred2(a, b) \leftarrow IsDaughter(b, a) \lor IsSon(b, a)$$
$$Pred3(a, b) \leftarrow \neg IsSon(b, a) \lor IsMother(b, a)$$
$$Pred4(a, b) \leftarrow IsFather(a, b) \land IsFather(a, b)$$
$$Pred5(a, b, c) \leftarrow \neg IsMother(a, b) \land IsMother(a, b)$$
$$Pred6(a, b) \leftarrow \neg IsSon(b, a) \land IsDaughter(b, a)$$
$$Pred7(a) \leftarrow \exists B, Pred1(a, B) \lor \exists B, Pred1(a, B)$$
$$Pred8(a, b) \leftarrow \exists C, Pred5(a, b, C) \lor \exists C, Pred5(a, b, C)$$
$$Pred9(a, b) \leftarrow \neg \exists C, Pred6(b, a, C) \land \exists C, Pred4(b, a, C)$$
$$Pred10(a, b, c) \leftarrow \neg Pred2(b, a) \lor Pred3(a, b)$$
$$Pred11(a, b) \leftarrow Pred8(a, b) \land Pred7(b, a)$$
$$Pred12(a, b, c) \leftarrow \neg Pred9(a, b) \lor Pred10(b, c, a)$$
$$Pred13(a, b) \leftarrow \neg Pred11(a, b) \land \forall C, Pred12(a, b, C)$$
$$IsUncle(a, b) \leftarrow Pred13(a, b) \land Pred13(a, b)$$

## C Task Description

### C.1 ILP

### C.1.1 Family Tree

For family tree tasks, they have the same initial predicates: $IsFather(X, Y)$, $IsMother(X, Y)$, $IsSon(X, Y)$ and $IsDaughter(X, Y)$. $IsFather(X, Y)$ is $True$ when $Y$ is $X$'s father. The other three predicates have the similar meaning.

- HasFather: $HasFather(X)$ is $True$ when $X$ has father. It can be expressed by:

$$HasFather(X) \leftarrow \exists Y, IsFather(X, Y)$$

- HasSister: $HasSister(X)$ is $True$ when $X$ has at least one sister. It can be expressed by:

$$HasSister(X) \leftarrow \exists Y, IsSister(X, Y)$$
$$IsSister(X, Y) \leftarrow \exists Z, (IsDaughter(Z, Y) \land IsMother(X, Z))$$

- IsGrandparent: $IsGrandparent(X, Y)$ is $True$ when $Y$ is $X$'s grandparent. It can be expressed by:

$$IsGrandparent(X, Y) \leftarrow \exists Z, ((IsSon(Y, Z) \land IsFather(X, Z))$$
$$\lor (IsDaughter(Y, Z) \land IsMother(X, Z)))$$

- IsUncle: $IsUncle(X, Y)$ is $True$ when $Y$ is $X$'s uncle. It can be expressed by:

$$IsUncle(X, Y) \leftarrow \exists Z, ((IsMother(X, Z) \land IsBrother(Z, Y)))$$
$$\lor (IsFather(X, Z) \land IsBrother(Z, Y))$$
$$IsBrother(X, Y) \leftarrow \exists Z, ((IsSon(Z, Y) \land IsSon(Z, X))$$
$$\lor (IsSon(Z, Y) \land IsDaughter(Z, X)))$$

- IsMGUncle: $IsMGUncle(X, Y)$ is $True$ when $Y$ is $X$'s maternal great uncle. It can be expressed by:

$$IsMGUncle(X, Y) \leftarrow \exists Z, (IsMother(X, Z) \land IsUncle(Z, Y))$$

### C.1.2 Graph Reasoning

For graph tasks, $HasEdge$ task have the same initial predicates: $HasEdge(X, Y)$. $HasEdge(X, Y)$ is $True$ when there is an undirected edge between node $X$ and node $Y$.

- AdjacentToRed: $AdjacentToRed(X)$ is $True$ if node $X$ has an edge with a red node. In this task, it also use $Colors(X, Y)$ as another initial predicate besides $HasEdge(X, Y)$. $Color(X, Y)$ is $True$ when the color of node $X$ is $Y$. It can be expressed by:

$$AdjacentToRed(X) \leftarrow \exists Y, (HasEdge(X, Y) \land Color(Y, red))$$

- 4-Connectivity: $4\text{-}Connectivity(X, Y)$ is $True$ if there exists a path between node $X$ and node $Y$ within 4 edges. It can be expressed by:

$$4\text{-}Connectivity(X, Y) \leftarrow \exists Z, (HasEdge(X, Y) \lor$$
$$Distance2(X, Y) \lor (Distance2(X, Z) \land HasEdge(Z, Y)) \lor$$
$$(Distance2(X, Z) \land Distance2(Z, Y)))$$
$$Distance2(X, Y) \leftarrow \exists Z, (HasEdge(X, Z) \land HasEdge(Z, Y))$$

- 6-Connectivity: $6\text{-}Connectivity(X, Y)$ is $True$ if there exists a path between node $X$ and node $Y$ within 6 edges. It can be expressed by:

$$6\text{-}Connectivity(X, Y) \leftarrow \exists Z, (HasEdge(X, Y) \lor$$
$$Distance2(X, Y) \lor Distance3(X, Y) \lor$$
$$(Distance2(X, Z) \land Distance2(Z, Y)) \lor$$
$$(Distance3(X, Z) \land Distance2(Z, Y)) \lor$$
$$(Distance3(X, Z) \land Distance3(Z, Y)))$$
$$Distance2(X, Y) \leftarrow \exists Z, (HasEdge(X, Z) \land HasEdge(Z, Y))$$
$$Distance3(X, Y) \leftarrow \exists Z, (HasEdge(X, Z) \land Distance2(Z, Y))$$

- 1-Outdegree: *1-Outdegree*$(X)$ is $True$ if there the outdegree of node $X$ is exactly 1. It can be expressed by:

$$1\text{-}Outdegree(X) \leftarrow \exists Y, \forall Z, (HasEdge(X,Y)$$
$$\wedge \neg HasEdge(X,Z))$$

- 2-Outdegree: *2-Outdegree*$(X)$ is $True$ if there the outdegree of node $X$ is exactly 2. It can be expressed by:

$$2\text{-}Outdegree(X) \leftarrow \exists Y, \forall Z, K(HasEdge(X,Y)$$
$$\wedge \neg HasEdge(X,Z) \wedge \neg HasEdge(X,K))$$

## C.2   RL

### C.2.1   NLRL Tasks

For the three NLRL tasks, *Unstack*, *Stack*, and *On*, the agent is trained to move the blocks to reach a certain configuration.

The action is represented by a binary predicate $Move(X,Y)$, which indicates moving block $X$ on block (or floor) $Y$. This action can be executed if it is legal, i.e., $X$ is not the ground, block $X$ has no blocks on it, and the same for $Y$ if it is a block. The three NLRL tasks share the same initial predicates $IsFloor(X)$, $Top(X)$, and $On(X,Y)$. Besides, there is one additional predicate $OnGoal(X,Y)$ for the *On* task only. $IsFloor(X)$ is $True$ when $X$ is the floor. $Top(X)$ is $True$ when block $X$ has no blocks on it. $On(X,Y)$ is $True$ when block $X$ is on $Y$. $OnGoal(X,Y)$ indicates that the goal for the *On* task is to move block $X$ onto block $Y$.

- Unstack: In this task, the goal is to move all the blocks on the floor. A policy that solves this task is:

$$Move(X,Y) \leftarrow IsFloor(Y) \wedge Pred(X)$$
$$Pred(X) \leftarrow Pred2(X) \wedge Top(X)$$
$$Pred2(X) \leftarrow \exists Y, Z, (On(X,Y) \wedge On(Y,Z))$$

  This solution is actually learned by DLM, where Predicates $Pred(X)$ and $Pred2(X)$ are invented during training. $Pred2(X)$ indicates that block $X$ is on top of another block, i.e., $X$ is not directly on the floor. $Pred(X)$ indicates that block $X$ is on top of a column of blocks and is not directly on the floor.

- Stack: The goal in this task is to stack all the blocks into one column. A policy for this task can be expressed as:

$$Move(X,Y) \leftarrow Pred(Y) \wedge Pred4(X,Y)$$
$$Pred4(X,Y) \leftarrow Pred3(X) \wedge Pred2(Y,X)$$
$$Pred2(X,Y) \leftarrow \exists Z, (On(X,Z) \wedge Top(Y))$$
$$Pred3(X) \leftarrow On(X,Y) \wedge IsFloor(Y)$$
$$Pred(X) \leftarrow On(X,Y) \wedge Pred2(Y,X)$$

  This solution was actually found by DLM where Predicates $Pred(X)$, $Pred2(X,Y)$, $Pred3(X)$, and $Pred4(X)$ are invented during training. Predicate $Pred(X)$ here has the same meaning as the one invented in *Unstack* task. $Pred2(X,Y)$ indicates that $X$ is a block and block $Y$ is has no blocks on it. $Pred3(X)$ indicates that block $X$ is directly on the floor. $Pred4(X,Y)$ indicates that block $X$ is directly on the floor and there is no block on it, and $Y$ is a block.

- On: The goal is to reach a configuration indicated by *OnGoal*. A policy for the *On* task can be expressed by:

$$Move(X,Y) \leftarrow (OnGoal(X,Y) \vee IsFloor(Y)) \wedge \neg On(X,Y) \wedge Top(X)$$

### C.2.2   Path

This is a single-source and single-target path finding problem in an undirected graph. For a given graph, the algorithm need to find if there is at least one path from the source node to the target node within $d$ steps.

The initial predicates are $Adjacent(X, Y)$, $IsStart(X)$ and $IsTarget(X)$. The graph is described as an adjacency matrix and $Adjacent(X, Y)$ is $True$ when node $X$ and node $Y$ are connected by an undirected edge. $IsStart(X)$ (resp. $IsTarget(X)$) is $True$ when node $X$ is the source (resp. target) node.

The action predicate is $GoTo(X)$. If $True$, it moves from the current node to the next node $X$.

A working policy for $Path$, requiring recursive predicates, can be expressed by:

$$GoTo(Y) \leftarrow \exists X, IsStart(X) \wedge Adjacent(X, Y) \wedge$$
$$(IsTarget(Y) \vee (HasEdge(Y, Z) \wedge IsTarget(Z)))$$
$$HasEdge(X, Y) \leftarrow Adjacent(X, Y) \vee$$
$$(\exists Z, HasEdge(X, Z) \wedge HasEdge(Z, Y))$$

### C.2.3   Sorting

In this problem, the algorithm needs to sort an array $a$ of $m$ integers into ascending order, by swapping those integers iteratively. Each index of array $a$ is treated as a constant.

The initial predicates are $SmallerIndex(X, Y)$, $SameIndex(X, Y)$, $GreaterIndex(X, Y)$, $SmallerValue(X, Y)$, $SameValue(X, Y)$, and $GreaterValue(X, Y)$. The first three describe index relations and the last three describe value relations. For example, $SmallerIndex(X, Y)$ is $True$ when $X < Y$, and $SmallerValue(X, Y)$ is $True$ when $a[X] < a[Y]$.

The action predicate is $Swap(X, Y)$. If $True$, it swaps $a[X]$ and $a[Y]$. Therefore, we have $m \times (m - 1)$ available actions.

The difficulty of the problem comes from the fact that the number of performed swap operations is limited. Hence, if the number of integers is small, a working policy for $Sorting$ can be expressed by:

$$Swap(X, Y) \leftarrow GreaterValue(X, Y) \wedge SmallerIndex(X, Y).$$

However, the general solution for any $m$ cannot be expressed without recursive predicates.

### C.2.4   Blocksworld

Recall that in this problem, there are two worlds: the source world where blocks can be moved and the target world that describes the desired block configurations. Each constant (block or ground) has 4 properties: a world ID, an object ID, an X coordinate, and Y coordinate. The ground has a fixed position $(0, 0)$.

The initial predicates are $SameWorldID(X, Y)$, $SmallerWorldID(X, Y)$, $LargerWorldID(X, Y)$, $SameID(X, Y)$, $SmallerID(X, Y)$, $LargerID(X, Y)$, $Left(X, Y)$, $SameX(X, Y)$, $Right(X, Y)$, $Below(X, Y)$, $SameY(X, Y)$, and $Above(X, Y)$. The first three compare two constants' world IDs, while the next three compare object IDs. The last ones compare two constants' X or Y coordinates.

The action predicate is *Move(X, Y)*. If true, it moves block $X$ onto $Y$ in the source world if it is a legal operation. It can be implemented as follows (found by a human expert):

$$Move(X,Y) \leftarrow PlanA(X,Y) \vee PlanBOnlyIf(X,Y)$$
$$PlanBOnlyIf(X,Y) \leftarrow PlanB(X,Y) \wedge \neg PlanAWork()$$
$$PlanAWork() \leftarrow \exists X, Y, PlanA(X,Y)$$
$$PlanB(X,Y) \leftarrow ShouldMove(X) \wedge NoGoodY(X) \wedge IsGround(Y)$$
$$\wedge InitialWorld(Y)$$
$$PlanA(X,Y) \leftarrow ShouldMove(X) \wedge ShouldMoveOn(X,Y)$$
$$NoGoodY(X) \leftarrow \neg \exists Y, ShouldMoveOn(X,Y)$$
$$ShouldMoveOn(X,Y) \leftarrow WellPlaced(Y) \wedge Clear(Y)$$
$$\wedge UnderBlock(X,Y)$$
$$ShouldMove(X) \leftarrow InitialWorld(X) \wedge Moveable(X) \wedge$$
$$\neg WellPlaced(X)$$
$$UnderBlock(X,Y) \leftarrow \exists Z, (Target(X,Z) \wedge UnderBlockTI(Z,Y))$$
$$WellPlaced(X) \leftarrow Matched(X) \wedge \neg HaveUnmatchedBelow(X)$$
$$UnderBlockTI(X,Y) \leftarrow \exists Z, (Target(Y,Z)$$
$$\wedge SameXDirectlyAbove(X,Z))$$
$$HaveUnmatchedBelow(X) \leftarrow \exists Y, (SameXAbove(X,Y) \wedge$$
$$\neg Matched(Y))$$
$$SameXDirectlyAbove(X,Y) \leftarrow SameX(X,Y)$$
$$\wedge DirectlyAbove(X,Y)$$
$$Moveable(X) \leftarrow Clear(X) \wedge \neg IsGround(X)$$
$$DirectlyAbove(X,Y) \leftarrow Above(X,Y) \wedge \neg \exists Z, Between(X,Z,Y)$$
$$Matched(X) \leftarrow \exists Y, Match(X,Y)$$
$$Clear(X) \leftarrow \neg \exists Y, SameXAbove(Y,X)$$
$$Between(X,Y,Z) \leftarrow Above(X,Y) \wedge Above(Y,Z)$$
$$Target(X,Y) \leftarrow SmallerWorldID(X,Y) \wedge SameID(X,Y)$$
$$Match(X,Y) \leftarrow \neg SameWorldID(X,Y) \wedge SameID(X,Y)$$
$$\wedge SameX(X,Y) \wedge SameY(X,Y)$$
$$InitialWorld(X) \leftarrow \neg \exists Y, SmallerWorldID(Y,X)$$
$$SameXAbove(X,Y) \leftarrow SameWorldID(X,Y) \wedge SameX(X,Y)$$
$$\wedge Above(X,Y)$$
$$IsGround(X) \leftarrow \neg \exists Y, Below(Y,X)$$

# D Experimental Set-Up

## D.1 Computer Specifications

The experiments are ran by one CPU thread and one GPU unit on the computer with specifications shown in Table 6.

Table 6: Computer specification.

| Attribute | Specification |
|---|---|
| CPU | $2 \times$ Intel(R) Xeon(R) CPU E5-2678 v3 |
| Threads | 48 |
| Memory | 64GB ($4 \times$16GB 2666) |
| GPU | $4 \times$ GeForce GTX 1080 Ti |

### D.2 Curriculum Learning

Every 10 epochs, we test the performance of the agent over 100 instances with a deterministic policy and a stochastic policy. If one of them reaches 100% then it can move to the next lesson. Our agents are trained only on one lesson at a time.

In the NLRL tasks, curriculum learning is not needed: the number of blocks during training is always 4.

In *Path*, the first lesson starts with 3 objects and finish with 10. In *Sorting*, we start with 3 objects and finish with 15. In *Blocksworld*, we start with 2 blocks (6 objects) and finish with 12 blocks (26 objects). In those 3 domains, the decreasing of the temperature and noise to obtain an interpretable policy is only applied during the last lesson.

For DLM-incr, curriculum learning is not needed: the number of blocks during training is always 7. The teacher produces the trajectories to learn. A new DLM is stacked only if the extracted formula is better than the previous one. If it is not, the parameters of the last DLM are reinitialized.

### D.3 Hyperparameters

#### D.3.1 Hyperparameters for dNL-ILP

For dNL-ILP, we train each task with at most $80,000$ iterations. Moreover, at each iteration, we use a new family tree or graph as training data, which is randomly generated from the same data generator in NLM and DLM, as backgrounds for training the model.
For task *HasFather*, *IsGrandparent* and *AdjacentToRed*, dNL-ILP can achieve 100% accuracy without learning any auxiliary predicates. For other ILP tasks, it has to learn at least one auxiliary predicate to induct the target. In practice, the performance decrease with increasing number of auxiliary predicates or variables, therefore here we only use at most one auxiliary predicate and at most three variables. Table 7 is the notions for all the hyperparameters for testing dNL-ILP. Table 8 shows hyperparameters for defining rules in dNL-ILP that achieve the best performance.

Table 7: Notions for hyperparameters used in testing Payani & Fekri (2019a)'s work.

| Hyperparameter | Explanation |
|:---:|:---:|
| $N_{arg}$ | The number of arguments |
| $N_{var}$ | The number of variables |
| $N_{terms}$ | The number of terms |
| $F_{am}$ | amalgamate function |
| $N_{train}$ | The number of nodes for training |
| $T$ | The number of forward chain |
| $N_{filter}$ | The number of tests for rules filter |
| $lr$ | Learning rate |
| $N_{epoch}$ | Maximum number of epochs for training model |
| $N_{iter}$ | The number of iterations for one epoch |
| $M_{terms}$ | Maximum number of terms in each clause |
| $\theta_{mean}$ | Fast convergence total loss threshold MEAN |
| $\theta_{max}$ | Fast convergence total loss threshold MAX |
| $\beta_1$ | ADAM $\beta_1$ |
| $\beta_2$ | ADAM $\beta_2$ |
| $\epsilon$ | ADAM $\epsilon$ |

Moreover, we use the same $N_{train}$ from NLM to train dNL-ILP (i.e. We set $N_{train} = 20$ for *HasFather*, *HasSister*, *IsGrandparent*, *IsUncle* and *IsMGUncle*. And we set $N_{train} = 10$ for *AdjacentToRed*, *4-Connectivity*, *6-Connectivity*, *1-Outdegree* and *2-Outdegree*). Other hyperparameters, such as hyperparameters for optimizer, remain the same for all tasks and consistent with Payani et al.'s github code

Table 8: Hyperparameters for defining and learning dNL-ILP rules for each task.

| Task | T | Auxiliary | | | | Target | | | |
|---|---|---|---|---|---|---|---|---|---|
| | | $N_{arg}$ | $N_{var}$ | $N_{terms}$ | $F_{am}$ | $N_{arg}$ | $N_{var}$ | $N_{terms}$ | $F_{am}$ |
| HasFather | 1 | – | – | – | – | 1 | 1 | 2 | or |
| HasSister | 2 | 2 | 1 | 1 | eq | 1 | 1 | 1 | max |
| IsGrandparent | 1 | – | – | – | – | 2 | 1 | 4 | eq |
| IsUncle | 7 | 2 | 1 | 3 | or | 2 | 1 | 4 | eq |
| IsMGUncle | 7 | 2 | 1 | 1 | or | 2 | 2 | 2 | or |
| AdjacentToRed | 2 | – | – | – | – | 1 | 1 | 2 | eq |
| 4-Connectivity | 7 | 2 | 1 | 1 | or | 2 | 1 | 5 | eq |
| 6-Connectivity | 7 | 2 | 1 | 1 | or | 2 | 2 | 7 | eq |
| 1-Outdegree | 3 | – | – | – | – | 1 | 2 | 2 | eq |
| 2-Outdegree | 3 | 2 | 0 | 1 | eq | 1 | 3 | 2 | eq |

Table 9: Hyperparameters for training dNL-ILP.

| $N_{epoch}$ | $N_{filter}$ | $N_{iter}$ | $M_{terms}$ | $\theta_{mean}$ | $\theta_{max}$ | $\beta_1$ | $\beta_2$ | $\epsilon$ |
|---|---|---|---|---|---|---|---|---|
| 400 | 3 | 200 | 6 | 0.5 | 0.5 | 0.9 | 0.99 | 1e-6 |

(https://github.com/apayani/ILP). For Family Tree tasks, we set $lr = 0.01$ when the loss from the last step is greater than 2, otherwise we set $lr = 0.005$. For Graph tasks, we set $lr = 0.01$. Other hyperparameters are shown in Table9.

### D.3.2 Hyperparameters for DLM

We have used ADAM with learning rate of 0.005, 5 trajectories, with a clip of 0.2 in the PPO loss, $\lambda = 0.9$ in GAE and a value function clipping of 0.2. For the softmax over the action distribution, we used a temperature of 0.01.

Table 10: Hyperparameters of the noise in DLM.

|  |  | Starting from | Exponential decay | Approximate final value |
|---|---|---|---|---|
| SL | temperature $\tau$ of Gumbel dist. | 1 | 0.995 | 0.5 |
|  | scale $\beta$ of Gumbel dist. | 1 | 0.98 | 0.005 |
|  | dropout probability | 0.1 | 0.98 | 0.0005 |
| RL | temperature $\tau$ of Gumbel distr. | 1 | 0.995 | task-dependent |
|  | scale $\beta$ of Gumbel dist. | 0.1 | 0.98 | task-dependent |
|  | dropout probability | 0.01 | 0.98 | task-dependent |

Table 11: Architectures for DLM on the different ILP and RL tasks.

|  |  | Depth | Breadth | $n_O$ | $n_A$ | IO residual[1] |
|---|---|---|---|---|---|---|
| Family Tree | *HasFather* | 5 | 3 | 8 | 2 | |
|  | *HasSister* | 5 | 3 | 8 | 2 | |
|  | *IsGrandparent* | 5 | 3 | 8 | 2 | |
|  | *IsUncle* | 5 | 3 | 8 | 2 | |
|  | *IsMGUncle* | 9 | 3 | 8 | 2 | |
| Graph Reasoning | *AdjacentToRed* | 5 | 3 | 8 | 2 | |
|  | *4-Connectivity* | 5 | 3 | 8 | 2 | |
|  | *6-Connectivity* | 9 | 3 | 8 | 2 | |
|  | *1-OutDegree* | 5 | 3 | 8 | 2 | |
|  | *2-OutDegree* | 7 | 4 | 8 | 2 | |
| NLRL Tasks | *Unstack* | 4 | 2 | 8 | 2 | |
|  | *Stack* | 4 | 2 | 8 | 2 | |
|  | *On* | 4 | 2 | 8 | 2 | |
| General Algorithm | *Sorting* | 4 | 3 | 8 | 2 | |
|  | *Path* | 8 | 3 | 8 | 2 | |
|  | *Path* (DLM-incr) | 6 | 3 | 8 | 3 | |
| Blocks World | *Blocksworld* (nIDLM) | 8 | 2 | 8 | 2 | ✓ |
|  | *Blocksworld* (imitation) | 8 | 2 | 8 | 2 | |

[1] Input-Output residual connections: As in NLM, all the input predicates of the DLM are given as input of every module. Similarly, every output of each module is given to the final predicate of the DLM.

