# OpenReview forum: "Differentiable Logic Machines"
_TMLR — Accepted by TMLR_

### Review · Reviewer_mWf5 · 2022-12-20

**Summary Of Contributions:**

The authors introduce Differentiable Logic Machines (DLMs), a new neuro-symbolic architecture for inductive logic programming-like tasks.  DLM build on a pre-existing architecture, namely Neural Logic Machines (NLMs), which feature a reduced number of parameters compared to other differentiable approaches to ILP.  SPecifically, DLMs take NLMs and a) Replace MLP layers with parameterized "soft" logic layers, b) Add more general "pooling" (or "predicate scrambling") operations between layers.  These additions are accompanied by improved training procedures for supervised learning and a new actor-critic approach for reinforcement learning problems.  These improvements are evaluated on an extensive set of experiments and ablation studies.

**Audience:**

Yes

**Broader Impact Concerns:**

None.  DLMs have the same potential negative impact as existing approaches (bar scaling to bigger problems).

**Claims And Evidence:**

Yes

**Requested Changes:**

A very minor change: the best results in the tables should be highlighted somehow (preferably in bold), if applicable.  This would help with readability.

**Strengths And Weaknesses:**

Strengths

+ The text is very well written and (surprisingly) mostly accessible to ILP non-experts.  All ideas are conveyed clearly.
+ Taken individually, all methodological contributions are reasonable.
+ Taken jointly, the proposed contributions bring improved interpretability (compared to NLMs) and scalability/success rate (compared to other differentiable ILP strategies)
+ The experimental setup is quite extensive and looks sound.
+ Results look very promising.

Weaknesses

- Taken individually, the methodological contributions are a bit incremental.  This is definitely not a problem for me.

All in all, this is a very solid piece of work, and I recommend acceptance.  As a matter of fact, I do not have any major changes to suggest!

---

> ### Author Response · Authors · 2023-02-08
> **Answer to Reviewer mWf5**
>
> We would like to thank you for the time you spent reviewing our work and for acknowledging its overall quality.
>
> We have applied the proposed modifications of the tables in the final version.

---

### Review · Reviewer_5Js5 · 2023-01-13

**Summary Of Contributions:**

The learning architecture involves a "continuous relaxation" of logic
programs (actually only a very restricted type of logic program) where
one assigns "weights to predicates instead of rules". Essentially
rather than assigning only 0 (false) or 1 (true) to a ground atom,
they can have values in the interval [0,1] much as in fuzzy logic
(although these values are interpreted as probabilities not fuzzy
values). So a binary variable is relaxed to a continuous one in the
normal way.

The authors present a neural architecture for learning with this
representation and present learning results which compare favourably
to other neural ILP methods. Comparison is based not only on
e.g. rewards for RL tasks, but also in terms of "interpretrability",
which considers whether a logic program can be extracted from the
neural learning.



**Audience:**

Yes

**Broader Impact Concerns:**

I had no such concerns.

**Claims And Evidence:**

No

**Requested Changes:**

The authors need to extend their empirical evaluation beyond
neural ILP methods to other ILP methods. This is a requirement. Since their system compares
favourably to some other neural systems, if they could (at least
sometimes) out-perform some respectable non-neural ILP approaches that
would be interesting. Note that non-neural ILP systems simply output a
logic program, so they naturally score well for "interpretability".

It would be good to see comparisons on tasks using predicates with
higher arities. The authors have chosen tasks where the input
and target predicates are unary or binary (which keeps the size of the
tensor representation under control). And only occasionally do we get
arity-3 invented predicates. What happens when one attempts a task
with high-arity predicates?


**Strengths And Weaknesses:**

The authors explain well how their approach relates to
existing neural ILP methods. The empirical comparisons (both RL and
non-RL) to these existing methods are quite thorough. The authors are
admirably up front about eg the difficulties of RL learning (with
sparse rewards). The exploration of curriculum learning makes sense.

The paper is (mainly) clearly written and it is not hard work to
understand what the authors have done (and potentially replicate
it) and why. As can be seen below the problem with the paper is its narrow
scope: the authors are basically only concerned with neural ILP
approaches, rather than the ILP problem per se.

LIMITATIONS OF DLM
==================

In Appendix B the authors correctly note that "the only FOL formulas
not covered by DLM are the ones containing function symbols". But
excluding function symbols is a very significant restriction since one
then has only a finite number of ground atomic formulae and ends up
with, essentially, propositional logic (each ground atomic formula
being a proposition). It is, of course, true when each proposition is
a particular grounding of some predicate, this can be exploited to
allow compact formulas(*): in this setting, a universally quantified
statement is a very nice compact version of a very long conjunctive
formula. So learning Datalog programs (and generalisations of Datalog,
like DLM) is a valuable goal, but since logic programs containing
function symbols (including very simple list processing ones like
"append") cannot be learned by DLM, it cannot be true (as claimed)
that DLM is a SOTA method for ILP - since many ILP systems are not so
restricted. The authors should be clear that they have a more modest
aim: doing ILP (and RL) in this particular restricted setting.

Note that when function symbols are absent, the idea to
"propositionalize" an ILP problem has a long history going back to
the early 90s.

(*) DLM does not exploit this potential compaction. In DLM, each b-ary
predicate P is represented by a tensor of order b with shape [m,...,m]
where m is the number of constants.

PERFORMANCE ON ILP TASKS
========================


In the conclusion it is stated that the authors' approach (DLM)
"obtains state-of-the-art results for inductive logic programming
tasks". However when the reader inspects Appendix B in the hope of
finding empirical support for that statement we find this: "Since DLM
is a neural-logic architecture, we only compare with other
neural-logic approaches dILP, NLM and dNL-ILP." Why only compare to
other neural methods if the goal is to support the claim that DLM
"obtains SOTA results for ILP"? There is no justification for the
"Since" here. One should either also compare to non-neural ILP methods also
or retract this claim.

Moreover, the non-RL ILP tasks described in Appendices B and C look
easy. We have no function symbols. Quite a few of these tasks do not
require recursion. There is no "background knowledge" apart from the
initial ground atoms. Also from the description of Algorithm 2 it
appears we have no noise (i.e. I assume the ground atoms P^O and P^T
"describing an ILP task" do so correctly). It is true, on the other
hand, that, in a number of cases, to get the desired logic program
some predicate invention is required which not all ILP systems deal
with well.


SPECIFIC POINTS
===============

- We have the statement: "Since traditional ILP systems can not handle
noisy, uncertain or ambiguous data, they have been extended and
integrated into neural and differentiable frameworks." Note first that
DLM greatly restricts (not extends) which logic programs can be
learned (compared to trad systems). What exactly does "can not handle"
mean here? Almost all ILP systems have ways of dealing with noisy
data. Is the claim here then that they do not do so very well? Perhaps
this is true, but it would be good to see some justification of this.

- "Variables refer to unspecified constrants." In FOL, variables refer
to unspecified ground terms which may or may not be constants.

---

> ### Author Response · Authors · 2023-02-08
> **Answer to Reviewer 5Js5**
>
> We would like to thank you for the in-depth review of our work and your valuable feedback.
>
> In the conclusion, we meant that we obtained state-of-the-art results among the neuro-symbolic methods, as written in the abstract and in the main text before. We understand that in its current form, it may be ambiguous, we have therefore rephrased our conclusion to make it less ambiguous.
>
> Regarding traditional ILP approaches, it is difficult to produce a fair comparison since the search space would not be the same. As far as we know, those methods often rely on carefully hand-designed and task-specific templates [1, 2] which is less the case with DLM. According to [2], traditional methods are superior to neuro-symbolic ones only when the dataset is small and noise-free. Finally, note that our main focus in this paper is to improve differentiable ILP methods in the reinforcement learning setting, not to tackle specific issues in ILP (e.g., how to deal with functions). Also, in the RL setting, it is not clear how to extend traditional methods.
>
> The maximum arity we tried was 4 in the 2-OutDegree task (invented predicates can be up to arity 4).
> However, the inference time and memory usage grow exponentially w.r.t the arity so it would be difficult to scale much more (see Section 4.1 for the exact complexity).
>
> [1] Law, M., Russo, A., and Broda, K. Inductive learning of answer set programs from noisy examples. Advances in Cognitive Systems, 7:5776, 2018.
>
> [2] Glanois, C., Jiang, Z., Feng, X., Weng, P., Zimmer, M., Li, D., Liu, W. and Hao, J. Neuro-Symbolic Hierarchical Rule Induction. Proceedings of the 39th International Conference on Machine Learning, 2022.

---

### Review · Reviewer_VmLJ · 2023-03-01

**Summary Of Contributions:**

This paper introduces a neural logic architecture, called DLM (Differentiable Logic Machine). It is inspired by NLM (Neural Logic Machine), but differs from it by using interpretable modules instead of MLPs, and by considering two additional operations, negation and preservation, incurring a higher computational complexity tough. This architecture is considered both for supervised learning (SL) and reinforcement learning (RL). For SL, it performs a classic binary cross-entropy minimisation, with additional tricks to learn an interpretable solution (Gumble noise in the softmax, decreasing temperature, dropout, such that the learnt soft logic operators are as deterministic as possible). For RL, it adapts PPO, a novelty a neural-logic context being to introduce a critic (instead of relying on monte-carlo rollouts for a vanilla policy gradient). The proposed architecture is experimented in both SL and LR benchmarks, compared with state-of-the-art (SOTA) approaches, and overall is competitive with non-interpretable SOTA while providing an interpretable solution. An ablation study is also provided.

**Audience:**

Yes

**Claims And Evidence:**

Yes

**Requested Changes:**

It could make sense to address the above clarity questions to widen the audience of this paper.

In addition, below are a few typos/things to be corrected:
* Sec. 3.2, « $d^{\pi_\theta}$, the stationary distribution of the Markov chain induced by the policy », this is wrong, it is the (unnormalised) discounted occupancy measure, not the stationary distribution (the best would be to define it formally)
* Eq. (5), at least an expectation is missing
* Sec 4.2.2, $r_t$ is used both for the reward and the reset gate vector (and arity in some places too)
* Appx A.4, a few formating issues « Here is the pseudo-code for XX »
* Alg. 3: I think $\beta$ should be $\beta_i$ and $\rho$ should be $\rho_i$ (both are decreased other time). Please also clarify what exactly how $\hat{A}$ is estimated, and the loss for $v^\psi$.
* some Eqs are numbered, other not, it seems that not all numbered Eqs are referenced in the tex, please homogenise.

**Strengths And Weaknesses:**

*Disclaimer*: the reviewer is far from being an expert on neural logic architectures, or inductive logic programming more generally

Overall, this well written paper provides novel contributions (the proposed neural logic architecture, the introduction of the critic for the RL part in this context), and provides a quite thorough empirical evaluation, both in SL and RL setting, with comparison to what seems to be the current SOTA, as well as an ablation of the Gumbel/temperature/dropout/critic components, and also study the computational and memory complexity. The appendix provides additional details, notably HPs for the experiments (open-sourcing the code would be welcomed).

The following questions are mainly clarifications requests (mostly related to the initial disclaimer, probably), and follows the structure of the paper.

**Neural Logic Machine**

This part is a bit hard to follow (even if it’s not contributive, the proposed approach build upon it). If I understand well, a set of predicates are transformed into another set of predicates through an MLP, but there are additional operations (expansion, reduction, permutation), how does the gradient flow through them? And how are chosen the permutations (the set of all possible permutations? could be pretty huge)? What could help could be to provide a more detailed graph of what happens, and also the equations describing what a computational unit does (eg, for a vanilla MLP would be something like $Y = \sigma(AX+b)$).

**Differentiable Logic Machine: architecture**

Same questions basically, and here also a more detailed graph of the computation unit could help, as well as Eqs describing what it does (ack it’s mostly done in Eqs (7-8). To sum up the difference between NLM and DLM are (please correct/complete if wrong):
* adding negation and preservation operation (same question about gradient flow)
* replacing the MLP of NLM by a soft-and and soft-or of a (learnt) convex combination of the input predicates

**DLM: actor-critic training**
* The RL modelling (state, action, next state) is not very clear. The state is a set of constants and predicates, but the network only takes predicates as inputs (and indirectly constants because predicates take constants as inputs)? The action is a predicate, but then what is the next state? A set of constants and predicates, but how are they obtained from the initial state-action couple?
* It is stated that designing a critic is difficult, and the proposed architecture is quite complicated, relying on GRU (why do we need memory?). However, if one considers NLM, it transforms predicates (so states) into predicates (so actions) through an MLP. Why not just considering an MLP taking a predicate as input and outputing a scalar, what would not work with this approach?
* It is stated that « Once an architecture for the critic is defined, different actor-critic algorithms could be applied ». However, the proposed architecture models a value function, while most AC requires a Q-function. Would the proposed architecture extends to Q-functions as well (maybe concatenating predicates corresponding to both the state and the action as input)?
* footnote 2, what means converting states (so predicates?) into images?
* Regarding the actor, there is another good reason for not seeking a deterministic policy: both policy gradient and PPO relies on having a policy with full support (otherwise one could rely on deterministic policy gradient)

**Incremental Training**

The network size does not depend on the number of constants, but it does depend on the number of predicates. So, in incremental training, at each iteration the architecture of the network changes, correct?

**Experiment**
What about comparing also to non-differentiable methods if they can handle this kind of problems? Discussing also what are the pros and cons. Indeed, the related work part could be also extended (even if briefly) with non-differentiable methods (in contrast to differentiable ones). Also, the metric is a bit surprising (even if it seems to be used in other works). It seems a bit unfair to report the results of the best model over N seeds only (even though the PSS metrics provides partially the variability). Why not just providing the average results over seeds?

---

> ### Author Response · Authors · 2023-03-13
> **Answer to Reviewer VmLJ**
>
> We would like to thank you for your careful review of our work and for noting the quality of our writing and the extensiveness of our empirical evaluation. We will answer your clarification questions below:
>
> NLM:
> An NLM module indeed transforms a set of predicates to another set. The 3 operations expansion, reduction, permutation are simple differentiable operations (i.e., repeating over a new dimension for expansion, minimum and maximum for reduction, and reindexing for permutation). The set of possible permutations is not too huge because it mainly depends on the number of variable which is limited.
>
> To be more concrete, if we follow your notations $Y = \sigma(AX + b)$ and consider Example 2 (page 7) with m objects and with two initial predicates: On(X1, Y1) represented by a matrix U2 of shape (m, m) and Top(X1) represented by a vector U1 of length m. Assuming that Y represents an unary predicate, X would be composed of only 3 predicates represented by the concatenation of min(U1, -1), max(U1, -1) and U2. Note that no permutations are used because Y is a unary predicate. We provide more details on the operations in Appendix A.1 and the new Table 3.
>
> DLM architecture:
> We confirm the differences are correct, we also use Gumbel-softmax for the soft-or and soft-and.
> Both NLM and DLM are built using differentiable operations. Please see Algorithm 1 in the appendix for the details of the gradient flow for DLM.
>
> DLM actor-critic training:
> The network only takes as input the ground predicates (i.e., atoms, represented as tensors) containing their valuations over all the constants. The set of constants only determine how to interpret the components of a tensor and are therefore not provided as inputs directly. For instance, a state is defined by all the state atoms, e.g., keeping only the positive ones, On(a, b) and On(b, floor) could define a state. The action is defined by an action atom, for instance Move(a, floor). The policy predicts the truth values of the predicate Move over all the constants. Finishing our example, the resulting next state would be On(b, floor), On(a, floor).
>
> We need memory to have an expressive aggregation function to compute a value from a set of input tensors, which also allows our architecture to generalize to any number of objects.
> An MLP in NLM computes an output tensor given a set of input tensors componentwisely, which ensures that the MLP is independent of the number of constants. Since the shapes of the input and output tensors are the same, such an MLP cannot compute an aggregate value as needed for a critic. To compute an aggregate value with an MLP, this MLP would need to depend on the number of objects, which would make the architecture not generalizable. Now, if NLM were used as critics, to reduce the inputs to a scalar only the min-pooling operation would be used across objects, which leads to a too simple aggregation function.
>
> We believe it would not be too hard to extend the architecture to Q functions by predicting a tensor with the same number of actions instead of a scalar.
>
> In the footnote 2, we meant that the predicates can be converted into an image because of the nature of the environment (blocksworld). If a block is present at a specific position the associated pixel in the Cartesian space is activated.
>
> Incremental Training:
> It is correct, the number of initial predicates in every first layer is increasing.
>
> Experiment:
> Regarding non-differentiable approaches, as we mentioned in another review, it is difficult to produce a fair comparison since the search space would not be the same. As far as we know, those methods often rely on carefully hand-designed and task-specific templates which is less the case with DLM. This difficulty and the difference between these approaches may explain the limited comparative evaluation in past work on ILP. To the best of our knowledge, we are only aware of [1, 2]. According to [2], traditional methods are superior to neuro-symbolic ones only when the dataset is small and noise-free. Also, in the RL setting, it is not clear how to use non-differentiable methods.
>
> We reused the metrics of previous works mainly to avoid having to recode their methods as some do not have available implementations.
>
> We corrected the different typos and included the requested explanations (the correction are highlighted in blue).
> Note that the advantage is estimated with TD($\lambda$)/Generalized Advantage Estimation and the loss of the critic is a mean squared error as stated in Algorithm 3.
>
>
> [1] Law, M., Russo, A., and Broda, K. Inductive learning of answer set programs from noisy examples. Advances in Cognitive Systems, 7:5776, 2018.
>
> [2] Glanois, C., Jiang, Z., Feng, X., Weng, P., Zimmer, M., Li, D., Liu, W. and Hao, J. Neuro-Symbolic Hierarchical Rule Induction. Proceedings of the 39th International Conference on Machine Learning, 2022.

---

### Decision · Action_Editors · 2023-06-05

**Recommendation:** Accept as is

**Comment:**

Most of the reviewers were happy with the contributions although they agree they are not highly significant. Yet, the paper fits the TMLR requirements by providing a new method for solving ILPs with a neural approach that still provides explainability. The claims are supported by experiments and comparison to other neural methods. One reviewer regrets the lack of comparison to non-neural approach and insisted that this should appear in the paper. I think this is a valid point that the authors could easily address. Yet it is not preventing publication according to TMLR guidelines and other reviewers. Comparison to similar methods are convincing enough to them.

Some modifications were requested to improve clarity and this was addressed by the authors, and acknowledged by the reviewers.



**Audience:**

The audience is somehow limited but the neural-logic field is expending and is large enough to deserve publication

**Claims And Evidence:**

In general, reviewers are happy with the claims and the way they are supported in the paper. Yet one reviewer regrets the lack of comparison with standard non-neural methods.

As action editor, I also agree that the lack of comparison weakens the paper and I would strongly suggest that authors provide even a simple experiment that supports their claim that this neural method outperforms non-neural methods. I understand that the literature is providing evidence that the proposed method would outperform standard ones on large datasets, yet implementation details or small differences in problem definitions can lead to slightly different conclusions. Also, the proposed method seems to have increased complexity compared to other neural logic machines, which means that comparison with non-neural methods is worth on the basis of computation cost as well.

Yet, I believe this paper is worth being published and I recommend acceptance.